# Generic comparison of lumen nucleation and fusion in epithelial organoids with and without hydrostatic pressure

Linjie Lu [1,2,3,4,16], Kana Fuji[5,16], Tristan Guyomar[1,2,3,4], Michèle Lieb[1,2,3,4], Marie André[1,2,3,4], Sakurako Tanida[5,6,7], Makiko Nonomura [8], Tetsuya Hiraiwa [5,9,10], Yara Alcheikh[11], Siham Yennek[12], Heike Petzold [11], Cecilie Martin-Lemaitre[13,14], Anne Grapin-Botton [11,12 ✉], Alf Honigmann [11,13,14 ✉], Masaki Sano [5,15 ✉] & Daniel Riveline [1,2,3,4 ✉]

Many internal organs in the body harbor a fluid-filled lumen. Lumen nucleation and fusion have been reported as dependent on organ-type during organogenesis. In contrast, the physics of lumen suggests that force balance between luminal pressure and cell mechanics leads to generic rules. However, this hypothesis lacks experimental evidence. Here we compare lumen dynamics for three different systems (MDCK cysts, pancreatic spheres, and epiblast model) by using quantitative cell biology, microfabrication, and theory. We report that the initial cell number determines the maximum number of lumens but does not impact the steady state, which is a final single lumen. We show that lumen dynamics is determined by luminal hydrostatic pressure. We also use MDCK cysts to manipulate cell adhesion and lumen volume to successfully reproduce the fusion dynamics of pancreatic spheres and epiblasts. Our results reveal self-organisation rules of lumens with relevance for morphogenesis and tissue engineering.

Organogenesis relies on individual cells that proliferate and interact to self-organize. An interplay between the physical properties of cells, their tissue organization, gene expression, and molecular control sets the complex rules for morphogenesis[1,2]. The cells in different organs use common means of controlling cell division, cell volume and shape, cell rearrangement and migration to enable different shapes to emerge[3]. Basic physical parameters of cells, such as pressure differences or surface tension, are crucial to the process. Understanding the physics of organogenesis requires clarifying these generic rules applicable to different cell types.

We took a generic approach to study the dynamics of a central structure in organs, *i.e.*, the lumen. Although lumen formation occurs in different epithelial organ models, several mechanisms of lumen formation are conserved across multiple systems such as hollowing after cell

[1]Institut de Génétique et de Biologie Moléculaire et Cellulaire, Illkirch, France. [2]Université de Strasbourg, Illkirch, France. [3]Centre National de la Recherche Scientifique, Illkirch, France. [4]Institut National de la Santé et de la Recherche Médicale, Illkirch, France. [5]Universal Biology Institute, Graduate School of Science, The University of Tokyo, Tokyo, Japan. [6]Research Center for Advanced Science and Technology, The University of Tokyo, Tokyo, Japan. [7]Department of Aeronautics and Astronautics, Graduate School of Engineering, The University of Tokyo, Tokyo, Japan. [8]Department of Mathematical Information Engineering, College of Industrial Technology, Nihon University, Chiba, Japan. [9]Mechanobiology Institute, National University of Singapore, Singapore, Singapore. [10]Institute of Physics, Academia Sinica, Taipei, Taiwan. [11]Max Planck Institute of Molecular Cell Biology and Genetics, Dresden, Germany. [12]The Novo Nordisk Foundation Center for Stem Cell Biology, Copenhagen, Denmark. [13]Technische Universität Dresden, Biotechnologisches Zentrum, Center for Molecular and Cellular Bioengineering (CMCB), Dresden, Germany. [14]Cluster of Excellence Physics of Life, TU Dresden, Dresden, Germany. [15]Institute of Natural Sciences, School of Physics and Astronomy, Shanghai Jiao Tong University, Shanghai, China. [16]These authors contributed equally: Linjie Lu, Kana Fuji. ✉e-mail: botton@mpi-cbg.de; alf.honigmann@tu-dresden.de; sano.masaki@sjtu.edu.cn; riveline@unistra.fr

division[4–8] or by apoptosis leading to cavitation[7–9]. In addition, theory for the physics of lumen was proposed[10,11] and some physical mechanisms for its dynamics were reported experimentally[12,13]. A systematic comparison between cellular systems is lacking so far.

Here, we probed self-organization of lumens on in vitro models. We use epithelial organoids as paradigms for lumen dynamics with physiological relevance[14–19]. We report a universal trend in various model systems, *i.e.* an increase in the number of lumens as a function of time followed by a decrease due to fusion until they all reach a single lumen. We also show that epithelial-derived lumens nucleate either after cell division or upon cell contact whereas epiblasts form lumens when they reach a rosette stage of 10 cells. By contrast, fusion of lumens is dominated by increase in pressure for pancreatic and MDCK spheres, whereas epiblast lumens fuse by cell motion. We show that epiblast lumens have negligible hydrostatic pressure in contrast to MDCK and pancreatic spheres. These rules are substantiated with a numerical simulation reproducing cell dynamics and lumen appearance using a phase field approach and by theoretical arguments. To further test these mechanisms, we used MDCK cysts to manipulate adhesion and lumen volume and we successfully reproduce the fusion dynamics of pancreatic spheres and epiblasts.

## Results

### Distinct phases of lumen dynamics

To track the growth and morphology of organoids (Fig. 1), we designed a microwell-containing device optimized for cell imaging and cyst tracking by using soft lithography[20] (Fig. 1a and Materials and Methods). Single devices contained microwells of different diameters adjusted to the measured mean cell dimension of each cellular system and a constant height equal to the cell height (Supplementary Fig. 1). This allowed us to follow different initial cell numbers over time within the same experiment, *i.e.*, 1, 2, 3, 4, 8, 16 cells (Fig. 1c). We used an MDCK cell line which expressed markers for cell-cell junctions and for lumens (see Materials and Methods) and this allowed us to track the number of lumens as a function of time in three dimensions (Fig. 1b–e). We observed two phases, Phase I with an increase in the number of lumens during the first 24 h followed by Phase II with a decrease over time eventually reaching a single lumen. This suggests that cells formed new lumens over time and that these lumens underwent fusion irrespectively of their initial number. During Phase II, these lumens could also undergo fusion irrespectively of their initial number and cyst shape (Fig. 1f, g). In addition, larger initial cell numbers correlated to larger number of lumens, ranging from a peak of 1 lumen for 1 initial cell at 24 h to 6 lumens on average for 16 initial cells. They reached single lumen within 24 h (1 day) for 1 initial cell and 192 h (8 days) for 16 initial cells.

We further explored whether this biphasic behavior was conserved in other systems. We plated pancreatic cells freshly isolated from fetal mouse pancreases at 13.5 days of development in the micro-cavities with adjusted dimensions and we tracked the evolution of lumen number over time (Fig. 2a and b). We quantified these dynamics (Fig. 2b) and we found a biphasic trend similar to the MDCK system (Fig. 1e). However, the distribution ranged from a peak of 1 lumen for 1 initial cell at 16 h to 5 lumens on average for 16 initial cells. Lumens fused into a single lumen within 24 h and 48 h, respectively. The same experiment with epiblasts led to similar conclusions (Fig. 2c and d): the lumen numbers increased and then decreased, reaching single lumens, ranging from an average peak of 1 lumen to 4 lumens for initial cell numbers of 1 cell and 16 cells, respectively. Lumen fusion into a single lumen happened within 2 days and 3 days, respectively. Altogether all systems exhibited the same qualitative behavior, an increase in lumen number with increasing initial cell numbers, and an increase of lumens number followed by fusion leading to single lumens (Supplementary Fig. 2).

The three systems exhibited similar phases but with different timing. We hypothesized that this may be due to the different cell cycle lengths of the different cell types. We thus plotted the number of lumens per cell cycle as a function of initial cell number for each system (Fig. 3a). Remarkably, the curves were similar for MDCK cysts and pancreatic spheres, with an increase of 0.2 lumens per cell cycle per initial cell number. In contrast, the slope was 5 times smaller for epiblasts, suggesting differences in the nucleation mechanisms. Following the dynamics of MDCK cells we could see that lumens formed by two mechanisms (Fig. 3b and Movie 1). As described previously, cells nucleated a lumen in the middle of the cell-cell contact after cell division[6,21] (see time 1:40 top Fig. 3b and Supplementary Fig. 3a). In addition, we found that two cells formed a lumen when they adhered to each other (see time 2:00 bottom Fig. 3b and Movie 2). The time needed for lumen appearance was similar between both processes (Fig. 3c). Both mechanisms were also observed in pancreatic cells (Fig. 3d, Movies 3 and 4) with the same 2 h timescale typically required to nucleate a lumen (Fig. 3e). It is worth noting that the low number of lumens per cell cycle per initial cell number suggests that lumens are nucleated during this Phase I but also undergo fusion with other lumens. The mechanism of nucleation was in sharp contrast with the appearance of lumens in the epiblasts (Fig. 3f and g): the lumen nucleated only when a critical number of about 10 cells formed a rosette (see time 48 h in Fig. 3f and g and Supplementary Fig. 3b, c). This may explain the distinct dynamics in Phase I.

### Hydrostatic pressure driving lumen fusion in epithelial models

The decrease in lumen number seen over time in Phase II suggested that lumen disappeared by fusing together (Fig. 4a). Live imaging enabled us to observe and quantify lumen fusion. We plotted the lumen fusion per cell cycle as a function of initial cell number (Fig. 4a). Unlike for Phase I, the three systems exhibited different fusion rates. The fusion was the fastest in pancreatic spheres with a reduction of lumen at a rate of 0.1 per cell cycle per initial cell number, followed by the epiblasts with a reduction of 0.08 per cycle per initial cell number. In contrast, the decrease was about 3-fold slower and smaller for MDCK lumens fusions than for the other organoids. To further compare the systems, we followed the dynamics of fusion of cysts for initial conditions of 8 cells until cysts reached similar dimensions and cell numbers (Fig. 4b). MDCK cysts exhibited a striking dynamic: the nearest neighboring lumens coming to contact fused by breaking the cellular junctions separating them over 60 h (Fig. 4b and Movie 5). We quantified these dynamics by plotting the lumen index (LI) per lumen; LI quantifies the ratio between the luminal area and outer cyst area to capture the respective increase in lumen volume (Fig. 4c, see Materials and Methods). The LI of one lumen increased whereas the LI of the neighboring lumens decreased in the period preceding the fusion. This sequence of events was similar for pancreatic sphere fusion but with faster kinetics (Fig. 4b and Movie 6) as evidenced in the LI quantification (Fig. 4d). In contrast, when we tracked the fusion of epiblasts lumens (Fig. 4b and Movie 7) we did not see a significant increase in the LI as illustrated by a constant LI prior to fusion (Fig. 4e). The large LI (approximately 1) for pancreatic spheres may suggest that the luminal pressure is larger than the MDCK sphere which presents a lumen index of 0.3 which is again 3-fold higher than the epiblast LI (see Supplementary Fig. 4). This indicates that increase in luminal pressure is large and important in pancreatic and in MDCK spheres. If hydrostatic pressure is a driver of fusion, we reasoned that it must rip apart the adhesion between cells, separating two lumens. To gain insight into adherens junctions, we quantified the levels of E-cadherins levels at junctions (see Fig. 5a, b, and Supplementary Fig. 5c). The mean concentration of E-cadherin was much larger for MDCK spheres compared to the other systems, which suggests that adhesion force may counteract lumen fusion via luminal pressure in this system. This difference in adhesion was further supported by inflation experiments where the cell layer was more easily disrupted in epiblast than in MDCK spheres (Supplementary Fig. 5d). This feature can explain the lower slope for

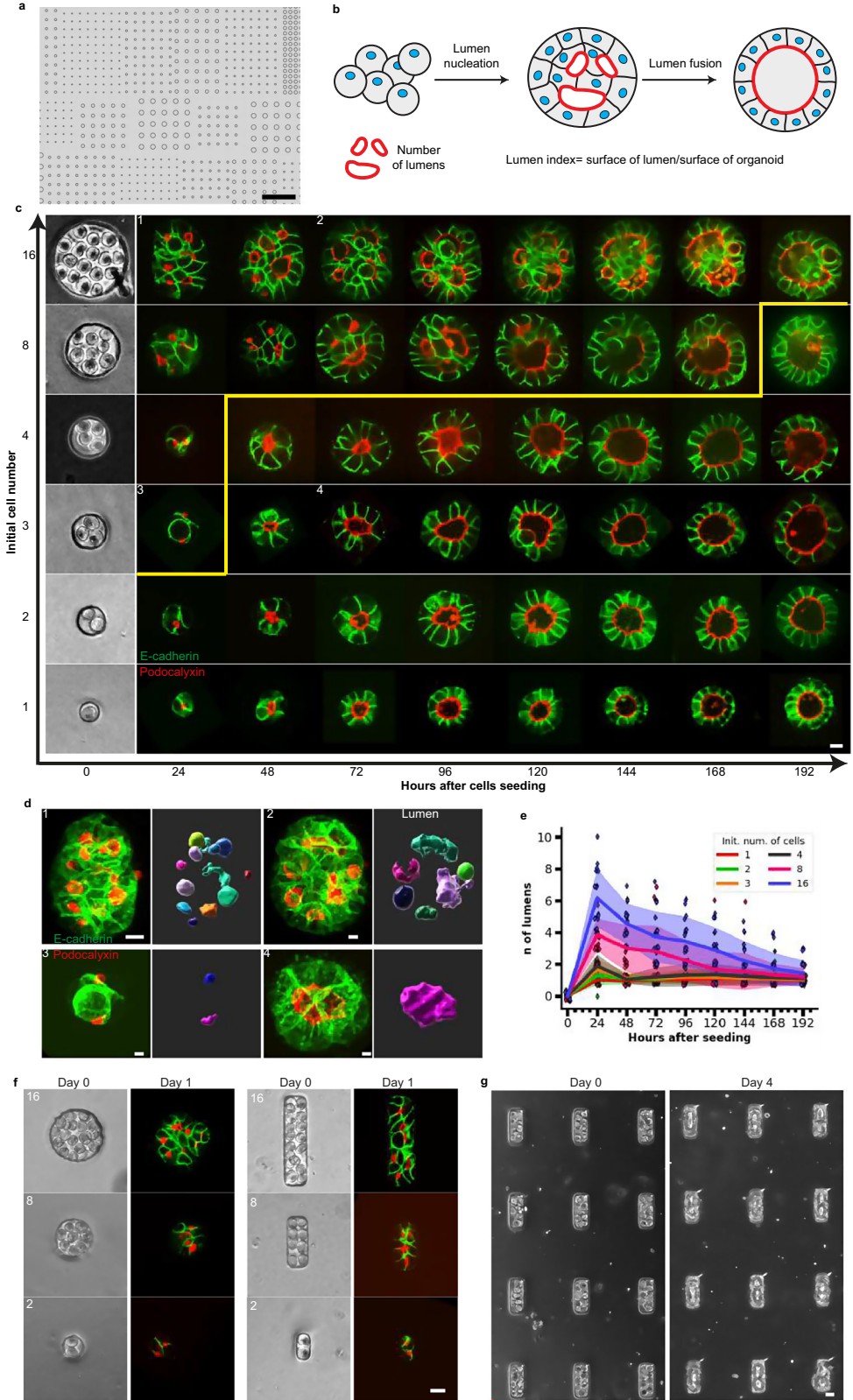

the lumen fusion for MDCK cysts compared to pancreatic spheres which may be dominated by large luminal pressure and low adhesion (Fig. 4a).

To test this central role for luminal hydrostatic pressure, we performed drainage experiments designed in our former study[22] (see Materials and Methods, Supplementary Fig. 6 and Fig. 5d–g). By cutting locally the cell layer with a laser, we could evaluate the luminal

hydrostatic pressure associated to each cyst. Our results show that MDCK cysts have a hydrostatic pressure of 65 Pa, larger than pancreatic spheres[23], in contrast to epiblast which have a hydrostatic pressure close to 0 Pa. We also tested that the mechanical properties of monolayers were similar (Supplementary Fig. 7), supporting the notion that luminal pressure was the driving force for nucleation and fusion of lumens.

**Fig. 1 | Lumen nucleation and fusion in MDCK spheroids with controlled initial cell numbers. a** Mask for the design of the cavity map. Cavities of different diameters were arrayed and designed to accommodate different initial cell numbers. Scale bar 500 μm. **b** Schematic representation of lumenogenesis. Dissociated plated cells adhere to each other, and then lumens (surrounded by red line) are nucleated and eventually fuse. **c** Typical dynamics of MDCK cysts forming from controlled initial cell numbers (E-cadherin in green, Podocalyxin in red). Initial cell numbers are shown at time 0 in cavities, and the shapes of the same spheroids are captured every day. The yellow line outlines the zone of MDCK cysts that reached the single lumen stage. Scale bar 10 μm ($N = 3$, $n \geq 10$). **d** 3D visualization of four lumens and spheroids corresponding to **c** and indicated as 1,2,3,4. left: 3D viewer of spheroids with E-cadherin in green and Podocalyxin in red. Right: 3 d viewer of

Lumen by Imaris. **e** Quantification of the number of lumens in MDCK spheroids as a function of time for different initial cell numbers: 1,2,3, 4, 8, 16 cells. Each point represents an individual cyst, while the dark line corresponds to the mean value, and the shaded region indicates the standard deviation. The number of lumens increases over time and with initial cell number. All cysts end up close to a single lumen. **f, g** External control of MDCK cyst growth using micro-well confinement. Comparison of lumen nucleation in circular and rectangular micro-wells shapes varying initial cell numbers. Podocalyxin is labeled in red, and E-cadherin is labeled in green. (**g**) Representative images show cyst morphology at Day 0 (left) and Day 4 (right) (3 independent experiments). Scale bars: 20 μm in f and in g. Source data are provided as a Source Data file.

## Motion-directed lumen fusion in epiblasts

Since the slope of lumen fusion was much lower for MDCK cysts than pancreatic spheres and epiblasts, we sought for an alternative mechanism driving lumen fusion. The lumen index value of epiblasts suggests that luminal pressure does not play a key role in the fusion. In particular the neighboring rosettes compacted into a sphere in epiblasts (Fig. 4b) and this was associated to the transformation of the outer layer of the cyst from an elongated shape to a sphere (Fig. 4i) in contrast to MDCK fusion case where the cyst remained spherical (Fig. 4f). We then measured the distance gained along the long axis of the cyst by cells: they corresponded to the distance needed for the lumens to fuse (Fig. 4j, k, see Materials and Methods). This is in sharp contrast with MDCK cysts (Fig. 4g, h). These experiments suggest that lumens fusion in epiblasts is driven by cells convergent directional motion associated with changes in cyst shape, whereas lumens fusions are mainly mediated by luminal pressure for MDCK cysts and pancreatic spheres.

## Toward quantitative generic rules for lumen dynamics

We turned to numerical simulations to reveal the rules in cell proliferation, cell adhesion, lumen formation, and luminal pressure, which could reproduce the main results across systems. These in silico experiments enabled us to test the hypotheses derived from our observations. In this context, we selected the phase field model, which captures the complex dynamics of cells and lumens once a proper free energy is set[24–26]. We assumed that cells grow and divide at threshold time and volume and form a lumen (Fig. 6a) and we modulate the increase in lumen volume and control cell-cell adhesion (see Methods and Supplementaryl Information). In silico experiments evolve spontaneously by setting the initial conditions as in the real experiments. We illustrate typical evolutions of the numerical cysts with 8 cells as initial cell number with the knowledge of the respective proliferation time of our systems, the same doubling time for MDCK and pancreatic spheres and stronger adhesion force for MDCK cells (Fig. 5). We obtained phenotypes similar to MDCK cysts (Fig. 6b blue, see also Movie 9); for larger luminal pressure with the same proliferation time (green), fusion looked similar to pancreatic spheres (Movie 10). Quantifications of lumen index in the numerical cysts for each case reproduced also the quantifications of experimental lumen index (compare Fig. 6d with Fig. 4c). Finally, to further validate the relevance of our simulations to experimental data, we quantified the dynamics in silico of the systems from initial stages by counting the number of lumens as a function of time. Strikingly, we could reproduce the biphasic behavior for all systems (Fig. 6c). These simulations support the importance of the interplay between cell proliferation, cell adhesion and luminal growth in setting quantitatively lumen dynamics across systems.

Based on our result of zero luminal pressure for epiblasts (Fig. 5d–g), we turned to alternative mechanisms. We analytically evaluated the number of cells needed to nucleate a lumen in the absence of hydrostatic pressure by assuming a similar free energy with the phase field model (Fig. 6e–h and Supplementary Information). Our

results show that 8–10 cells are needed to nucleate a lumen (see Fig. 6f, h) in good agreement with our results for epiblasts whereas 2 cells are sufficient for a cyst with luminal pressure (Figs. 6g, i, j). In the epiblast, in the absence of hydrostatic pressure, the apical side and lateral sides generate a total force that needs to encompass the resisting force associated to the cell monolayer and the matrix. As a consequence, more cells are needed to outcompete this threshold force. In contrast, high hydrostatic pressure spontaneously allows this opening of lumen already with 2 cells. This difference substantiates the differences between MDCK/pancreatic spheres on the one hand and epiblasts on the other hand.

## Testing the model in MDCK cysts and epiblasts

Our modeling and experimental observations suggested that lumen growth, adhesion between cells, and cellular properties governed the differences between the three systems. We further tested this experimentally using the MDCK system as a reference and perturbing these parameters. To evaluate the role of lumen growth in fusion, we prepared MDCK cysts with two lumens, and we designed an inflation experiment using a micro-pipette to inject fluid[22] (Fig. 7a and Supplementary Fig. 8). To show the fusion, we used dextran in the pipette. We could induce the fusion within minutes. This suggests that an increase in lumen growth rate can drive fusion of MDCK lumen faster than it normally takes, along the result of a faster response promoted by an increase of lumen volume in pancreatic spheres. In contrast, epiblast could not sustain this volume change upon injection, leading the whole structure to collapse instead of triggering fusion (Supplementary Fig. 9). The same enhanced fusion was obtained by an osmotic shock (Fig. 7b and c and Materials and Methods). In addition, the apparent role of cell-cell adhesion for MDCK cysts was further tested by simultaneously decreasing cadherin-mediated adhesion by chelating calcium with EDTA and using an anti E-cadherin blocking antibody[27,28] (Fig. 7d). We observed that lumen number is significantly decreased, suggesting that lumen fusion was facilitated (Fig. 7d, e). This suggests that adhesion between cells is an impediment to fusion in MDCK cysts. In addition, we tested the behavior of E-cadherin KO MDCK cell line for 8 cells and 16 cells initial conditions: the biphasic behavior was reproduced but this E-cadherin cell line presented a faster fusion (see Fig. 7f–h). This further confirms that adhesion between cells can prevent fusion. Finally, to test the impact of cellular properties on lumen formation, we used a MDCK cell line in which the tight junction proteins ZO1 and ZO2 were knocked-out[22,29]. This cell line was shown to have smaller LI due to increased apical contractility[22] and resembled the epiblast case with smaller lumen and elongated cells (see Supplementary Fig. 10a). We initiated the MDCK ZO-1/2 KO2 KO cysts with 1 to 16 cells and we repeated the observation of the number of lumens as a function of time and initial cell numbers (Supplementary Fig. 11a, b). The results show that fusion was facilitated in the ZO-1/2 KO cyst. Several features of this ZO-1/2 KO MDCK cyst corresponded with epiblasts cysts, such as low LI and similar mechanical response to inflation (Supplementary Fig. 10b), as well as facilitated fusion as a function of cell number. This suggests that tight

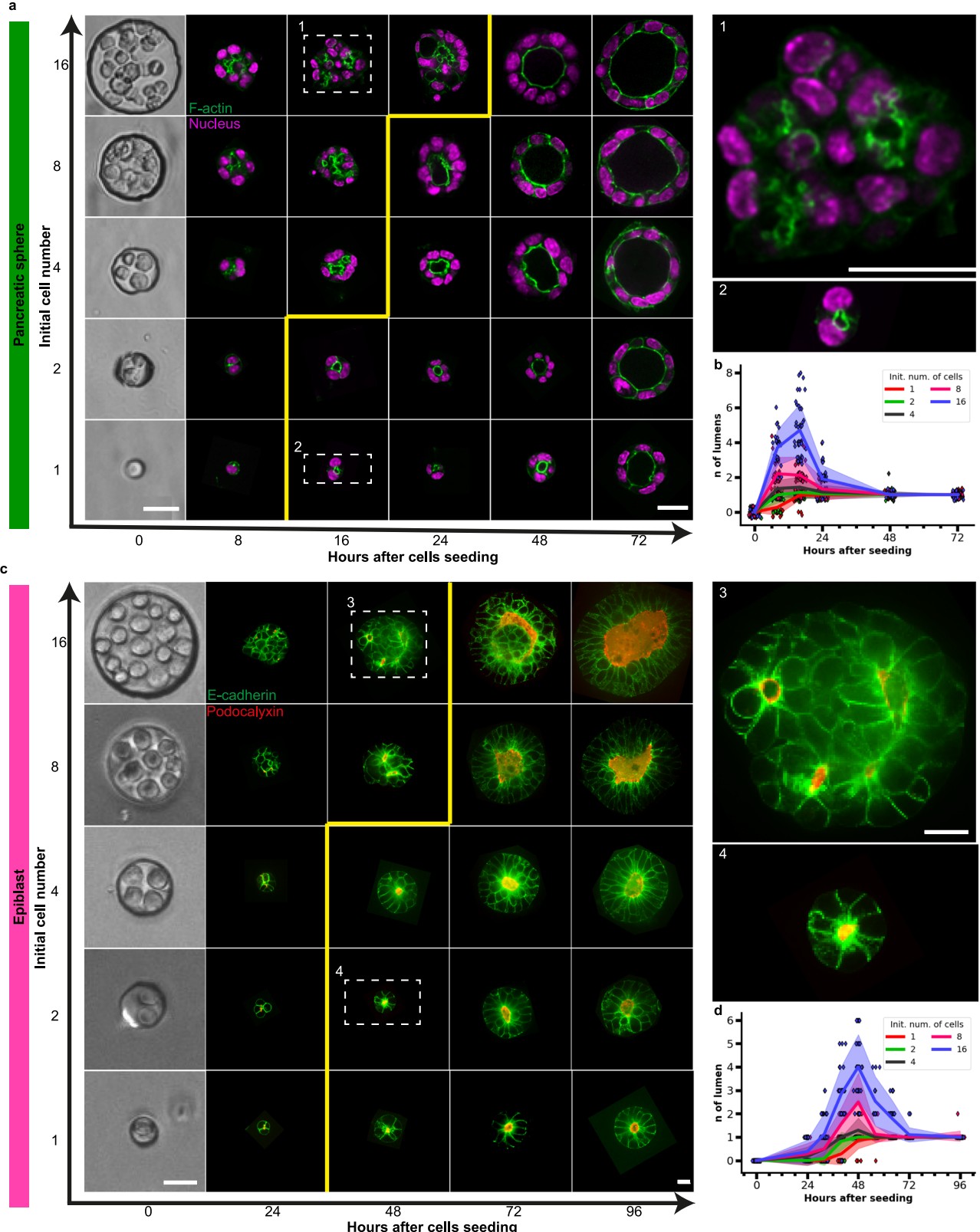

junction deletion contributes to faster fusion. These experiments of MDCK cysts transformations with mechanical or biological perturbations support the notion that lumen fusion generically results from this interplay between luminal pressure and mechanical cell interactions.

Finally, to show that activity was required for the fusion in the case of epiblasts, we recorded the fusion of epiblasts in the presence of the myosin inhibitor, blebbistatin (Fig. 8a). Fusion was prevented, which shows that active forces are needed to promote fusion – in contrast to MDCK cysts fusion (Supplementary Fig. 12). This was also obtained when adhesion was reduced in MDCK ZO-1/2 KO (Supplementary Fig. 11c). To further substantiate these fusion mechanisms with no pressure, we prepared numerical epiblasts similar to the experimental

**Fig. 2 | Lumen nucleation and fusion in pancreatic and in epiblast spheres.**
**a** Typical evolution of pancreatic spheres shapes as a function of initial cell number and time (F-actin in green, nucleus in Magenta) ($N = 3$, $n > 20$). Scale bars 10 μm at time 0 and at time 96 h with 3D zooms on outlined pancreatic spheres.
**b** Quantification of the number of lumens in pancreatic spheres as a function of time for different initial cell numbers: 1, 2, 4, 8, 16 cells. **c** Typical evolution of epiblasts as a function of initial cell number and time (E-cadherin in green, Podo-calyxin in red) ($N = 3$, $n > 20$). Scale bars 10 μm at time 0 and at time 72 h with 3D zooms on outlined epiblasts. **d** Quantification of the number of lumens in epiblast

system with two lumens and small luminal pressure (Movie 11) with an elongated configuration similar to the experimental case (compare Fig. 4b). Based on experimental measurements in this configuration (Fig. 4i–k), we imposed radial cell motion by adding active forces (Fig. 8b) and we successfully recapitulated the fusion (Fig. 8c) as well as cell motion within the cyst (Fig. 8d, e) and axis ratio of the cyst (Fig. 8f).

## Discussion

We have shown that the number of lumens depends on the initial cell number and their evolution exhibits similar biphasic behaviors for MDCK, pancreatic spheres and epiblasts (Fig. 9). The nucleation phase is dictated mainly by appearance of new lumens whereas the second phase is dominated by fusion to reach a single lumen for all systems. Nucleation mechanisms are shared between MDCK cells and pancreatic spheres, both when cells divide and when two cells adhere to each other, whereas epiblasts need about 10 cells to nucleate a lumen. MDCK and pancreatic spheres fusions are predominantly driven by an increase in lumen index and high pressure, whereas fusion is determined by active cell motion in epiblasts with low lumen index and negligible pressure. Our experimental perturbations of MDCK fusions suggest that luminal pressure in competition with cell-cell adhesion controls fusion. In pancreatic spheres the lumen index is larger than the MDCK cysts, and fusion is likely driven by luminal ion pumping, whereas epiblast fusions are dominated by cells motion.

The mechanism of lumen nucleation after cell division has been reported for MDCK cells[6,30,31]. It is consistent with the nucleation mechanism we report for MDCK, and we report it for pancreatic spheres. In addition, we also add the lumen nucleation associated with contact between cells as well, which supports the notion that several mechanisms of lumen nucleation could co-exist with similar timing. It is worth noting that lumen nucleation was reported in an assay between a cadherin-coated surface and a single cell[32], suggesting that adherens junctions formation between cells per se could trigger lumen formation. Finally, we report that a larger number of cells is needed to nucleate a lumen in epiblasts and it is consistent with Shahbazi et al.[33,34]. Despite these multiple mechanisms for lumen nucleation, we report and explain in a unified way the mechanisms for the nucleation, i.e. Phase I, across systems.

We distinguish fusion between lumens triggered by ion-pumping mechanisms leading to an effective increase in hydrostatic pressure of the lumen in competition with cell-cell adhesion and fusion between lumens triggered by cell re-organization. This difference can be understood by a simple force balance argument: when the lumen index is large, the hydrostatic pressure of the lumen pushes the cell apical side and the cell monolayer thereby competing with cell-cell contacts; in contrast, when lumen index is low, the hydrostatic pressure of the lumen is low and interactions between cells essentially determine the potential fusion between lumens. Lumen fusion by increased osmotic pressure was reported in various situations in vitro and in vivo[12,35,36], along our observations, and this illustrates that common rules of self-organization across systems could determine the morphogenesis of organs in 3D.

It is interesting to understand why epiblasts show very low hydrostatic pressure. We note that MDCK cysts and pancreatic

spheres are sealed in the sense that Dextran cannot enter the luminal space[22,23]. This is in sharp contrast with epiblasts[34,37]. We propose that this difference is associated with low luminal pressure. In addition, we conjecture that this property is associated with tight junction maturation, i.e. tight junctions are fully matured in MDCK cysts, whereas they do not reach this state in epiblasts. This proposition is consistent with the similar lumen dynamics exhibited by epiblasts and by ZO-1/2 KO MDCK cell lines. Future experiments will be needed to identify the molecular mechanisms associated with tight junction maturation.

The nucleation and growth of lumens are governed by the same rule in three systems, which is the balance between the luminal pressure and the contraction of the cell lateral walls and apical membrane, and this is consistent with the minimization of the total free energy. Epiblast organoids cannot spontaneously nucleate a lumen during cell division because of low lumenal pressure, which costs more free energy. In the process of minimizing free energy, the cell passes through a low free energy rosette state. This is achieved by using the active fluctuations (centripetal motion) of the cells. However, as explained in Fig. 6, even when the internal pressure is nearly 0 Pa, a rosette consisting of more than 10 cells is unstable and leads to the formation of a lumen that further lowers free energy. In MDCK cysts and pancreas spheres, lumens spontaneously nucleate between two cells due to a finite lumenal pressure. Lumen fusion process is also known as the result of free energy minimization under the influence of active fluctuations of cell motion. In this sense, these systems follow universal rules.

We propose that these nucleation and fusion mechanisms could be tested on other organoid systems. The timing of cell proliferation compared to lumen fusion dynamics could be tuned to optimize the target size of the organ with the relevant cell number and the number and size of lumens. Systems may select fast pumping, like the pancreas, to allow multiple lumens to fuse rapidly with potential change in luminal pressure to form a single duct[38]. By contrast, systems that would need to keep compartments such as the thyroid gland may have developed larger adhesion properties to prevent fusion and allow each zone to keep potential differences in composition[39].

In considering functional differences between cells connected to lumen dynamics, it is interesting to note that cells change their states in the case of epiblasts[40]. From stem cells, they exit from pluripotency and follow paths of differentiation, leading to the right localization in the final organs. We propose that this orchestration of proliferation with lumens nucleation and fusion could also be optimized to generate organs with the relevant shape and cell numbers, but also with the correct cell-state distributions. Future experiments coupling our approach with spatial transcriptomics will allow us to test this hypothesis.

Our results could shed light on the synthesis of artificial organs. Indeed, it was reported that cell printing was a promising method to generate organs[41,42]. Our results show that the cell number at plating, their growth rates, and their fusions contribute to the dynamics of the organ formation. This initial condition correlates with the morphology and functions of the organ. As a result, due care to the force balance would need to be evaluated in the synthesis of organs, and our

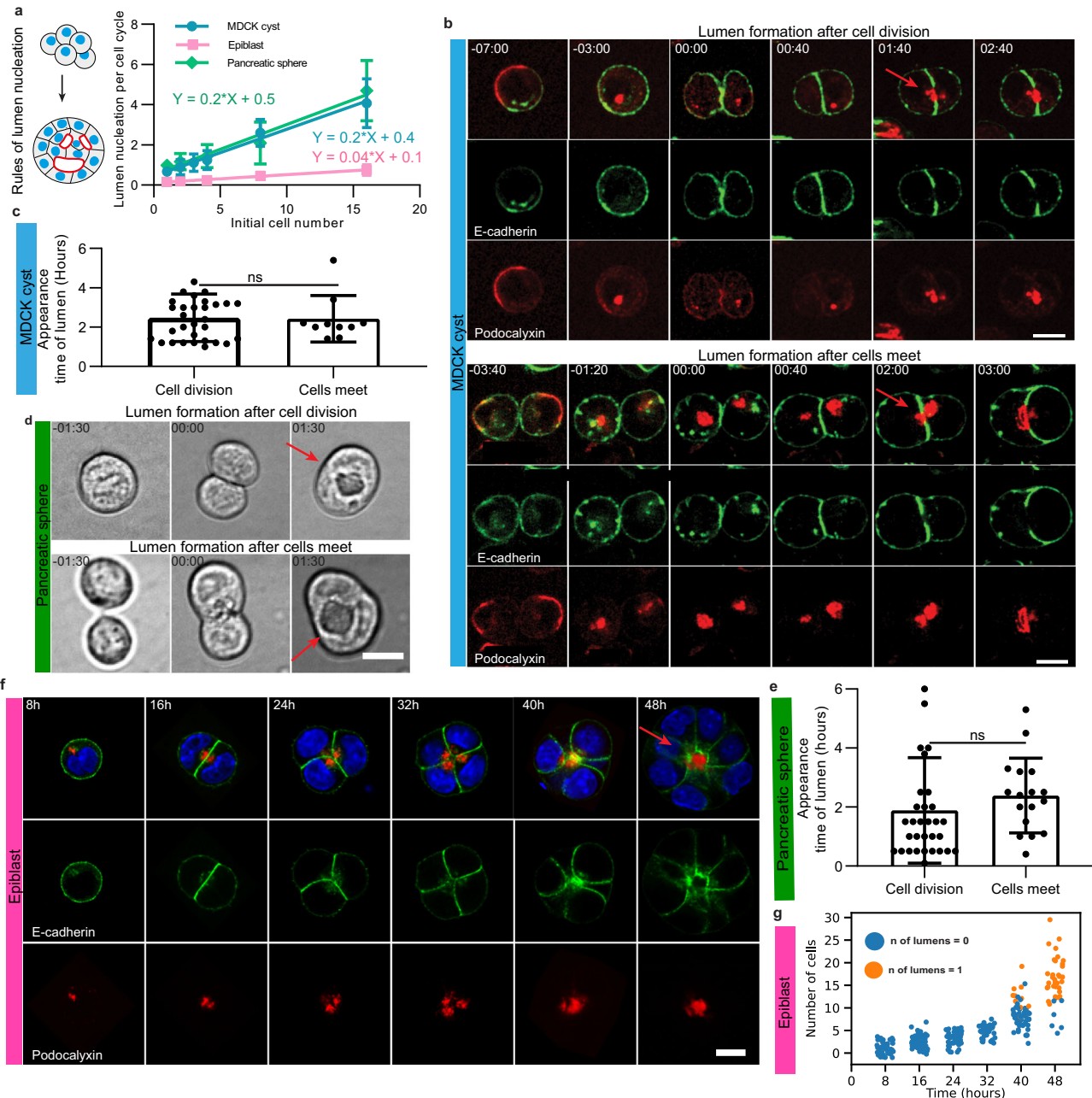

**Fig. 3 | Rules of lumen nucleation in the 3 systems.** The lumens are nucleated after cell division and after cells meet in MDCK spheres and pancreatic spheres, but they form after about 5 cell divisions in epiblasts. **a** Lumen nucleation per cell cycle as a function of initial cell numbers. The increase is linear for the 3 systems (mean value ± standard error of the mean and the curve represents the fit of the mean). MDCK and pancreatic spheres have the same slope, whereas epiblasts have a slope 5 times lower (For MDCK cyst: $N = 3$, $n \geq 10$. For pancreatic sphere and epiblast, $N = 3$, $n > 20$). **b** Snapshots of lumen nucleation in MDCK cysts. Top: Lumen formation after cell division in MDCK cysts (Movie 1). Bottom: Lumen formation after contacts between cells in MDCK cysts (Movie 2) (E-cadherin in green and Podocalyxin in red). Time relative to the junction formation set as time 0. Time in hh:mm. Scale bar 10 μm. **c** Quantification of lumen appearance time after cell division $N = 5$, $n = 32$ or after cells meet $N = 4$, $n = 10$. **d** Snapshots of lumen nucleation in pancreatic spheres, Top: lumen formation after cell division, see also Movie 3. Bottom: lumen formation after two cells meet; see also Movie 4. Time relative to the junction formation set as time 0. Time in hh:mm. Scale bar 10 μm. **e** Quantification of the lumen nucleation after cell division $N > 3$, $n = 32$, or after cells meet $N > 3$, $n = 17$. **f** Lumen nucleation in epiblasts fixed at 8 h, 16 h, 24 h, 32 h, 40 h, and 48 h after cell seeding, stained with DAPI in blue, podoxalyxin in red, and E-cadherin in green. **g** Number of cells as a function of time ($N = 2$, $n = 287$). For (**c**) and (**e**), Mean ± SD. Two-tailed with t-test. For (**a**, **c**, **e**, **g**), source data are provided as a Source Data file.

framework with its numerical simulations could serve as a solid basis to predict the future shape of the targeted organs.

## Methods

### Cell sources and expansion

We used 3 cellular systems, MDCK II cell lines, mouse embryonic cells (mES cells)[43], and pancreatic spheres[44]. Other mutant cell lines were used for MDCK: MDCK II E-cadherin-GFP/Podocalyxin-mScarlett/Halo-CAAX[45], MDCK II ZO1/2-KO[22], and MDCK II E-cadherin KO[46].

The MDCK II cell lines were cultured in MEM (Gibco 410900028) with 5 % Fetal Bovine Serum (Sigma, USA), 1 mM Sodium Pyruvate (Gibco 11360-070), and 1x NEAA (Gibco 11140050). MDCKII cells were resuspended every 2 to 3 days with trypsin-EDTA after they reached

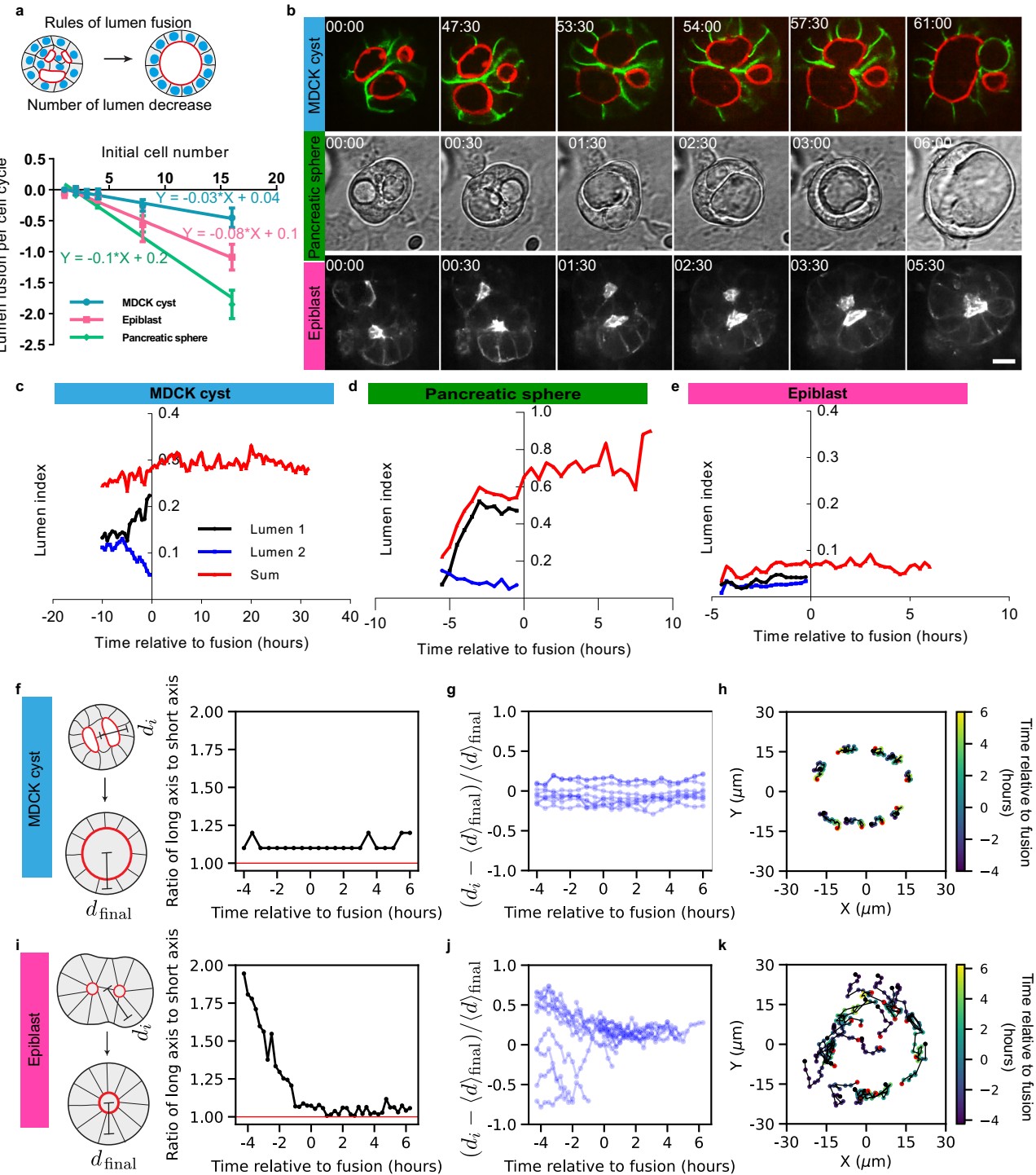

**Fig. 4 | Rules of lumen fusion in the 3 systems.** MDCK and pancreatic spheres lumens fuse by increasing the lumen index whereas epiblasts fuse by cell motion with low lumen index. **a** Top: scheme of lumen fusion corresponding to Phase II. Bottom: quantification for the speed of lumen fusion. The plot represents lumen fusion per cell cycle as a function of initial cells number (mean value ± standard error of the mean and the curve represent the fit of mean). Pancreatic spheres and epiblast display faster lumen fusion than MDCK cysts (For MDCK cyst: $N = 3$, $n \geq 10$. For pancreatic sphere and epiblast, $N = 3$, $n \geq 30$). **b** Lumen fusion across systems. Top: MDCK cyst (E-cadherin in green and Podocalyxin in red, also see Movie 5). Middle: Pancreatic sphere (phase contrast images, also see Movie 6). Bottom: epiblast (Sir-actin, also see movie 7). Comparison between lumen indices (ratio of

surface of lumen over the surface of cyst) for the three systems presented on panel **b** MDCK cyst (**c**), pancreatic sphere (**d**) and epiblast (**e**). For each system, the blue and black curves correspond to individual lumen 1 and 2 on panel (**b**).
**f–k** Characterization and comparison between cellular dynamics and tissue morphology in MDCK cyst (top) and epiblast (bottom). **f, i** Elongations of the cyst defined as the ratio of long axis over short axis of the organoid as a function of time. **g, h** Distance between the center of cells and the center of cysts over time. During the fusion process, MDCK cells are at a constant distance from the center of the cyst whereas epiblast cells move inwards. **h, k**. Single-cell trajectories. Time is indicated with color bar and red point indicates the last time point. Epiblast cells display centripetal motion. Source data are provided as a Source Data file.

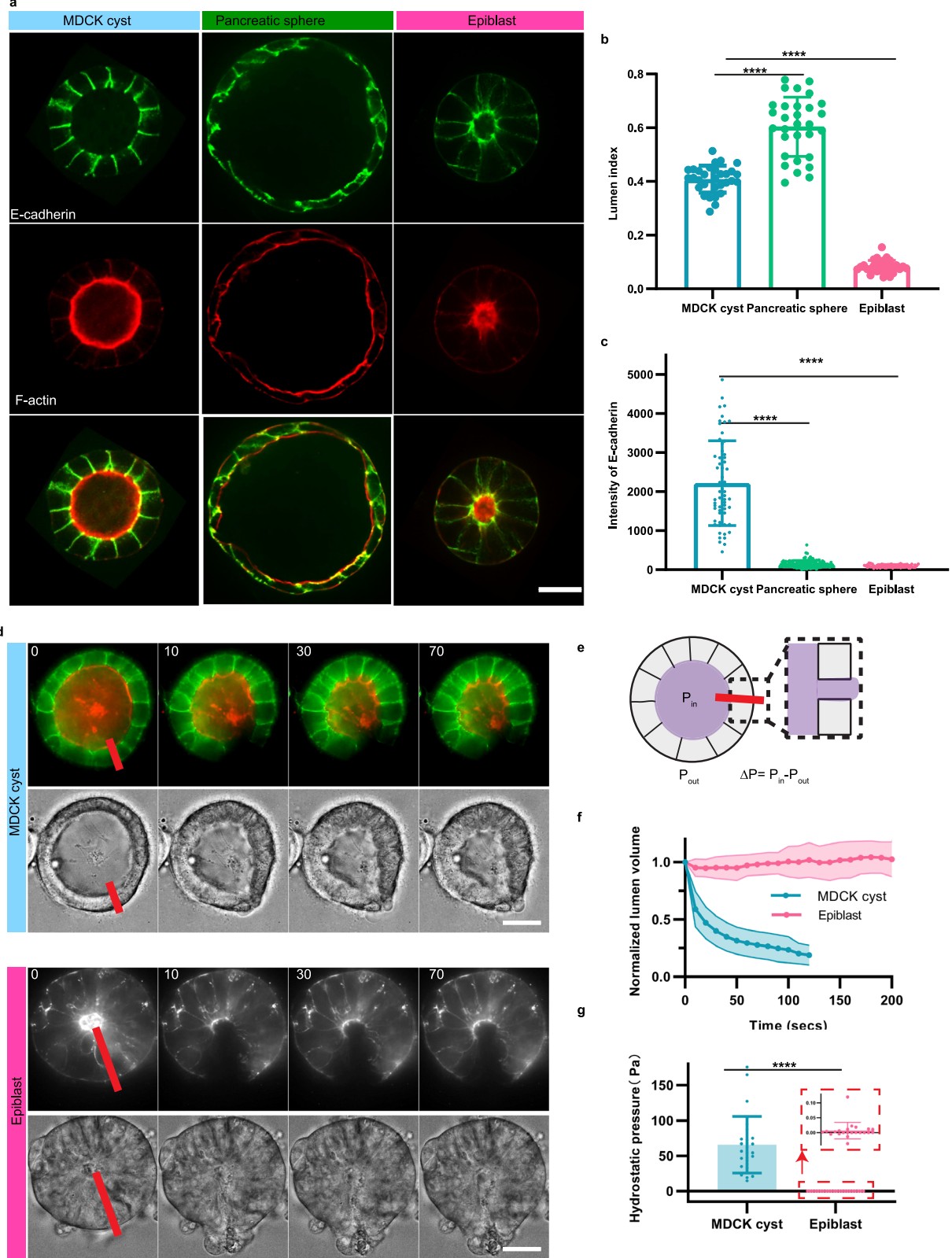

70-95% confluency[47]. A seeding density of about 3*10^5 cells per 75 cm^2 was used for sub-culture. R1 ES-cell line was used for the culture of epiblast model[43]. Mouse embryonic stem cells were expanded with 1:1 DMEM/F12 (ThermoFischer 31331028) and neurobasal medium (Gibco 21103049) supplemented with 1x N2 (Gibco 17502048), 1x B27 (Gibco 12587010) and 1x NEAA (Gibco 11140050), 55 μM 2-Mercaptoethanol (Gibco 21985023), 3 μM CHIR 99021 (Sigma SML 1046-5 mg), and 2x LIF

produced at IGBMC-Strasbourg in non-adhesive flasks[43]. Cells were sub-cultured every 4 days until the size of spheres reached a diameter of 80 μm. They should be called "epiblasts model" and we also use "epiblasts" in the text for simplicity. Pancreatic spheres were prepared from the dissection of E13.5 embryos (mouse CD1 from Charles River Laboratory) using the protocol reported in Greggio et al.[44] and used without passaging.

**Fig. 5 | Comparison between phenotypes and hydrostatic pressure in MDCK cysts, pancreatic spheres and epiblasts. a** Typical images of the 3 systems with readouts for E-cadherin in green and F-actin in red. Scale bar 10 μm. **b** Quantification of lumens index for the 3 systems ($N = 3$, $n > 20$). **c** Comparison of E-cadherin levels between the 3 systems defined as intensity of E-cadherin on the junction minus intensity in the cytoplasm and normalized by the intensity of the background ($N = 3$, $n > 60$). For (**b**) and (**c**), Mean ± SD and two-tailed with Mann Whitney test; **** corresponds to $p < 0.0001$. **d** Measurements of hydrostatic pressure in MDCK cysts and epiblasts by laser cutting. Images show the midplane

cross-sections of cysts before and after laser cutting, with red segments indicating the positions of cuts. **e** Estimation of hydrostatic pressure in the lumen using the Hagen-Poiseuille law (see Methods). **f** Lumen volume changes over time following laser cutting. The dark line represents the mean volume change, while the shaded region indicates the standard deviation. **g** Hydrostatic pressure measurements show an average of 65 Pa in MDCK cysts and close to 0 Pa in epiblasts. Mean ± SD, two-tailed with t-test, $p < 0.0001$. Data are based on three independent experiments ($N = 3$), with 27 MDCK cysts ($n = 27$) and 22 epiblast samples ($n = 22$). Scale bar: 10 μm. Source data are provided as a Source Data file.

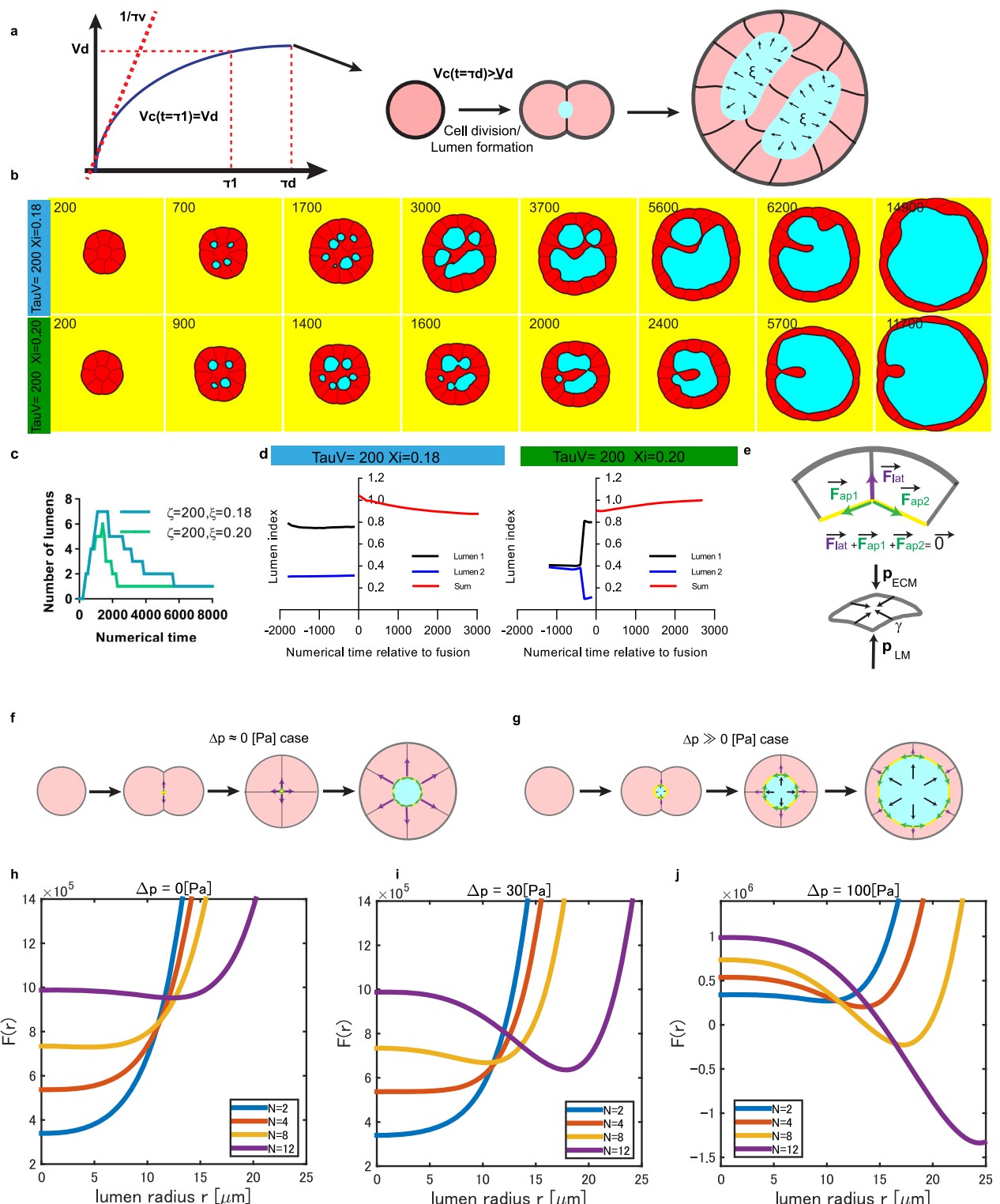

**Fig. 6 | Phase field model for the nucleation and fusion of lumens for the 3 systems. a** The cells increase in volume and divide when they reach a threshold volume or after a threshold time. A lumen is formed after each division. The numerical cysts grow over time; see Movie 8 for the typical numerical evolution of cysts. **b** The pressure and proliferation time are controlled, and two conditions are plotted as a function of numerical time. Each case corresponds qualitatively to each experimental system, see also Movie 9 and Movie 10. **c** Quantification of the number of lumens as a function of time. The number of lumens increases and then decreases. These two subsequent phases are similar to the experimental phases reported in Fig. 1 and in Fig. 2. **d** Numerical lumen indices were measured for the 2 conditions. Their dynamics are similar to the 2 experimental dynamics reported in Fig. 4. **e**–i **e** Top: schematic representation of the force balance at the vertex,

highlighting the lateral (lat) and apical forces (ap) (top). Bottom: the schematics depict the Young-Laplace law with γ the surface tension and p the pressure ($p_{ECM}$, the extracellular matrix pressure, and $p_{LM}$ luminal pressure). **f** Simplified schematic of the model for lumenogenesis in the absence of hydrostatic pressure. When Δp is close to 0 Pa, lumen formation requires a minimum threshold of 10 cells to nucleate a lumen. In contrast, when Δp>>0 Pa, 2 cells spontaneously nucleate a lumen (**g**). **h**–**j**. Free energy dependency as a function of lumen radius for distinct hydrostatic pressures. F(r, R_min): **h** Δp = 0 [Pa] for $N$ = 2, 4, 8, 12. **i** Δp = 30 [Pa] for N = 2, 4, 8, 12; and **j** Δp = 100 [Pa] for $N$ = 2, 4, 8, 12. Numerical parameters are $γ_A$ = 100 [Pa · μm], $γ_L$ = 100 [Pa · μm], $R_0$ = 20 [μm], L = 40 [μm], $L_0$ = 38.5 [μm], k = 0.02, $k_E$ = 0.01 for all cases. For $N$ = 12 case, we assumed a regular dodecahedron shape for the calculation. Source data are provided as a Source Data file or Github.

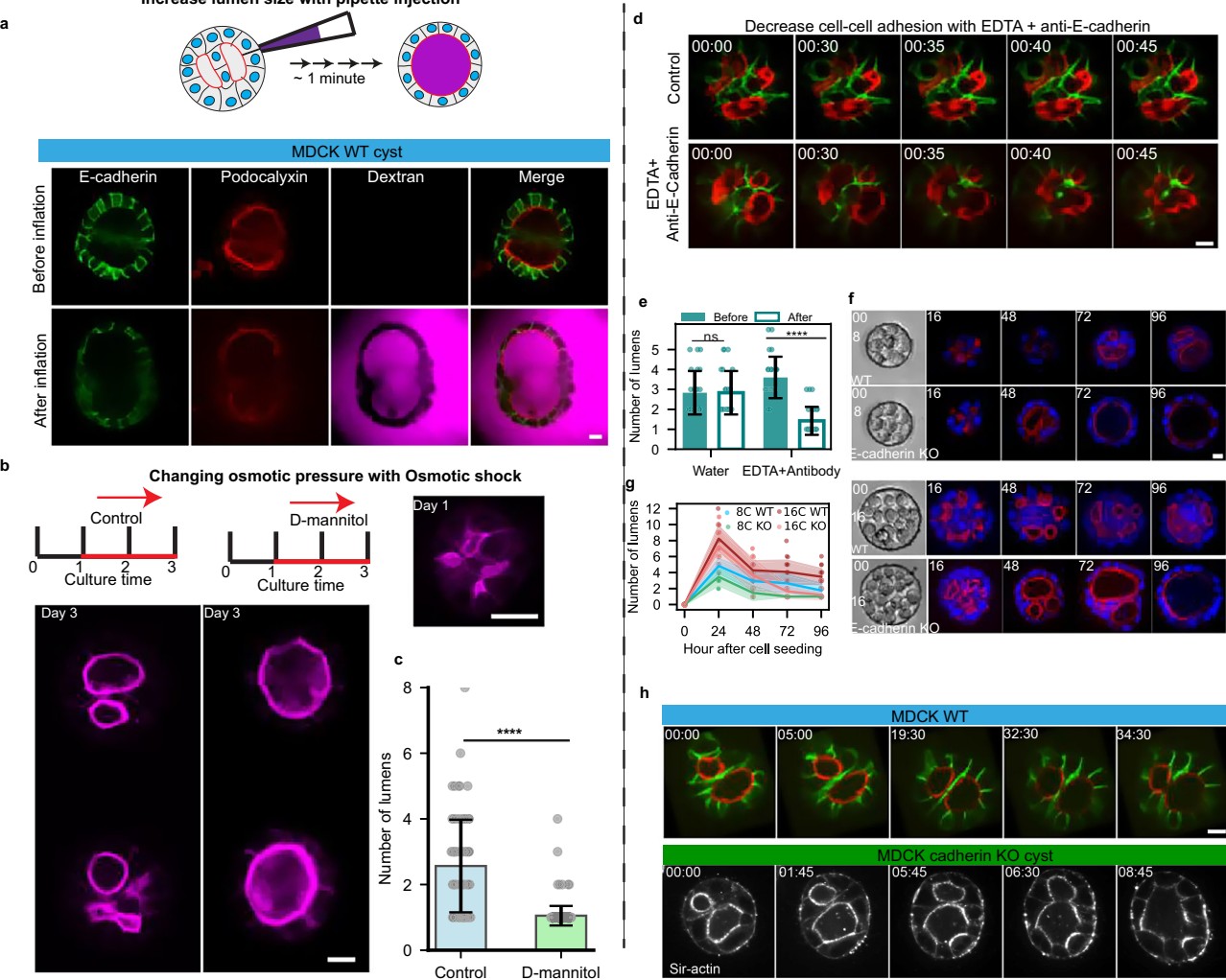

**Fig. 7 | Testing the fusion mechanisms with perturbation experiments on MDCK cysts.** Increasing lumen size by injection or decreasing cells adhesion leads to faster to lumen fusion. **a** Inflating MDCK spheres with micropipette. E-cadherin in green and podocalyxin in red and Dextran in Magenta. Scale bar: 20 μm. The inflation triggers the fusion between lumens. **b** Effects of osmotic shock on MDCK cysts with multiple lumens. Experimental setup (top left): MDCK cysts were cultured for 1 day before being treated under control conditions or with D-mannitol. The culture duration is indicated in days. Representative images of cysts are shown at two time points: day 1 (top right) and day 3 (bottom left) for each condition. Cysts are stained for F-actin (magenta) to visualize structural changes. Scale bar: 10 μm. **c** Statistical analysis based on three independent experiments (N = 3), with 79 cysts in the control group (n = 79) and 198 cysts in the D-mannitol-treated group (n = 198). p < 0.0001. Mean ± SD, two-tailed with Mann-Whitney

test. **d** Decrease in cells adhesion leads to faster lumen fusion. Snapshot of timelapse with control (top) and EDTA + antibody block (bottom). E-cadherin in green and podocalyxin in red. Scale bar: 10 μm. **e** Quantification of number of lumens after EDTA+antibody treatment. Two independent experiments. Statistical analyses: ****$p < 0.0001$; ns, not significant. (n control = 15; n EDTA = 16). Mean ± SD. Two-tailed with Wilcoxon matched-pairs signed rank test.
**f**–**h** Decreasing cell adhesion by using MDCK E-cadherin KO. The comparisons between MDCK WT and E-cadherin KO cysts over time and initial cell number in (**d**) (top two rows: 8 cells, bottom two rows: 16 cells). **g** Quantification of the number of lumens in WT and E-cadherin KO cysts as a function of time for the initial cell number 8 and 16 cells. The dark line corresponds to the mean value, and the shaded region indicates the standard deviation. Source data are provided as a Source Data file.

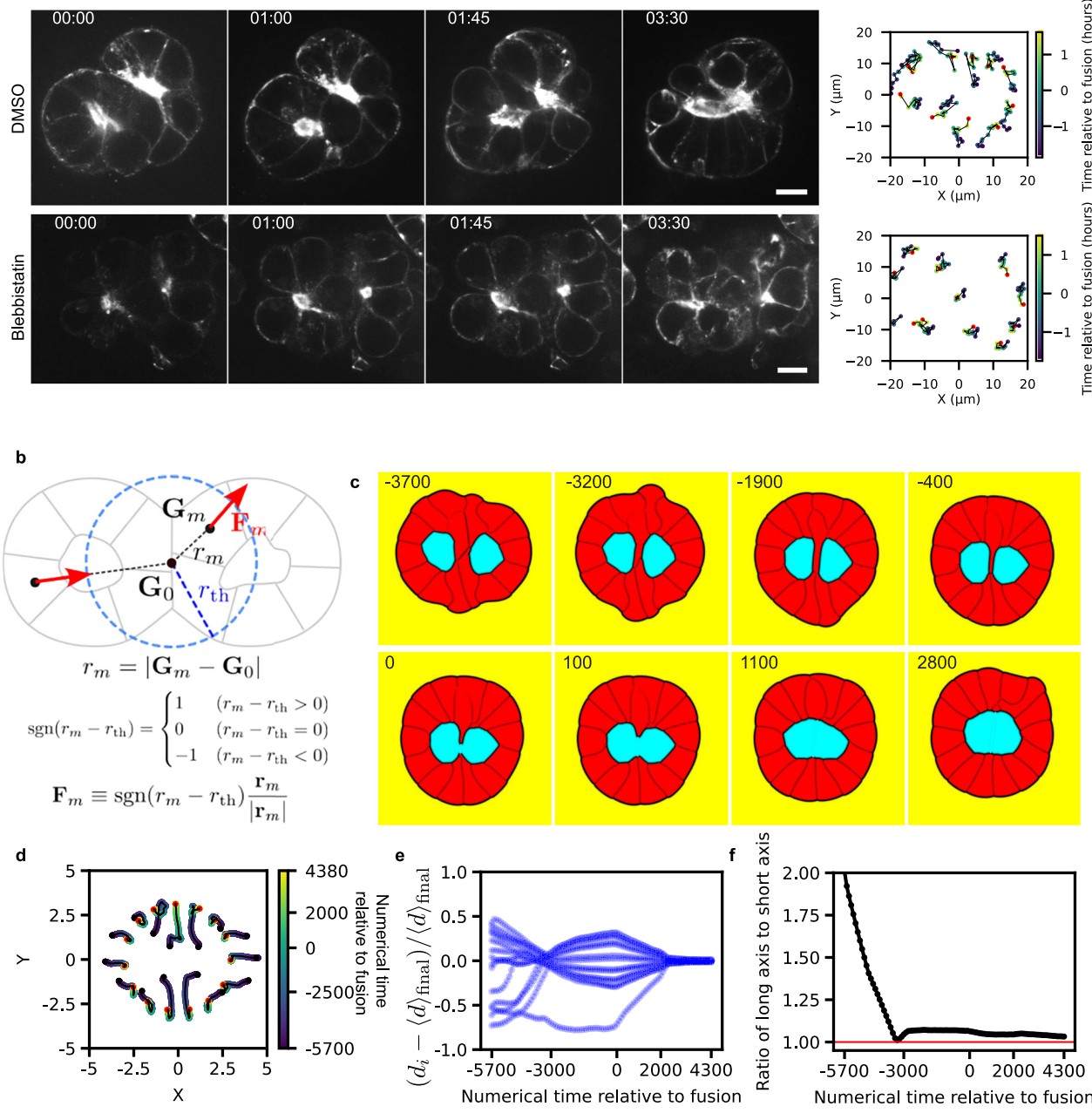

**Fig. 8 | Perturbation of lumen fusion in the epiblast. a** Representative images of the lumen fusion process in the epiblast. The top panel shows the control condition treated with DMSO, and the bottom panel shows the effect of blebbistatin treatment. Lumens are visualized with SiR-actin (gray). Right panel: Single-cell trajectories during the fusion process under DMSO treatment (top) and blebbistatin treatment (bottom). Time is indicated by the color bar, with the red point marking the last time point. Scale bar: 5 μm. **b**–**f** Numerical simulations for the fusion of epiblast-like cysts, see also Movie 11. **b** Definition of parameters for the fusion of epiblast-like cysts. **c** Fusion of two lumens in epiblast-like condition. **d** The cell motions are tracked as well as the distance of cells with respect to the center of mass of the system (**e**) and the change in aspect ratio of the cysts (**f**). The numerical dynamics correspond closely to the experimental time evolution of the same measurements in epiblasts fusion, see Fig. 4j–l. Source data are provided as a Source Data file.

## Cells diameter measurement

For the three systems, single cells were plated after trypsinisation and labeled using 10 nM SiR-actin (TEBU-BIO, 251SC001). The middle planes of spherical cells were imaged. The associated surfaces were measured with Fiji and the distributions of cells diameters were plotted (see Supplementary Fig. 1). Cavities diameters were designed accordingly by taking the mean value of each distribution to control the initial cell numbers (Supplementary Fig. 1c). All cavities had a cell height from 10 μm to 17 μm to keep the height similar to one cell diameter.

## Microfabrication and cavity map

We designed the samples with a map of patterns for cavities in order to: (i) track the evolution of the same cysts up to a week, (ii) test a large number of cysts with the same initial cell number and (iii) test the effects of different initial cell numbers for the same biological repeat (see Fig. 1a). The same strategy was adapted for each system by designing the cavity map accordingly.

Cavities were prepared using soft lithography as described in Bhat et al.[20]. Briefly, we designed a mask with AutoCad to obtain a large number of motifs and different diameters. The motifs were selected to

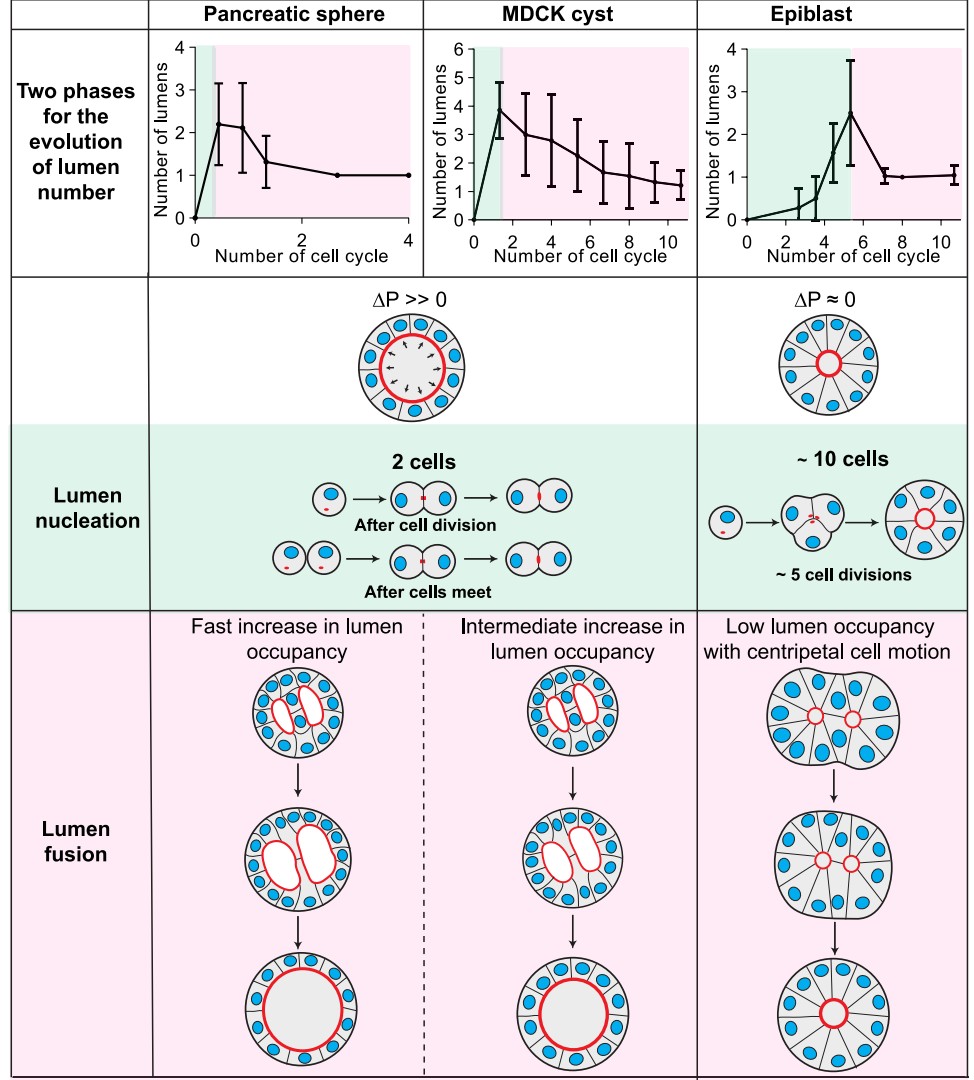

**Fig. 9 | Comparison of lumen nucleation and fusion in organoids with/without hydrostatic pressure. Row 1:** Pancreatic spheres, MDCK cysts and epiblasts have two phases for the number of lumens as a function of time (mean value ± standard error of the mean and curves represent fits, for details see Figs. 3a and 4a). **Row 2: Key differences in hydrostatic pressure observed between the three systems. Row 3:** Nucleation rules are similar for pancreatic spheres and MDCK cysts, lumens form after cell division and when two cells meet; in contrast, a minimum of about 10 cells is required for the lumen nucleation of lumens in epiblasts. **Row 4:** Lumens fusion are similar between pancreatic spheres and MDCK cysts but with distinct pumping rates. In pancreatic spheres, fast pumping leads to an increase in lumen index and subsequent fusion; in MDCK cysts, la ower increase in lumen index leads the fusion. In contrast, fusion occurs with centripetal cell motion in epiblasts with low lumen occupancies.

contain many initial cells number conditions. We used the following calculation of the motif's diameter, rescaled with the mean cell diameter: S (surface of cavities) = number of initial cells*surface of cells. These designs were printed on a photomask and then we transferred these patterns on SU-8 silicon wafer with soft lithography. Next, the design was replicated on a PDMS mold. Finally, these designs were transferred to cover-glass which allows us to achieve higher resolution images.

## Cell seeding in microfabricated cavities for the control of initial cell number

To generate cysts and organoids in micro-fabricated cavities, we seeded cells in micro-cavities with the following steps based on our former protocol[20,48]. Briefly, (i) the microfabricated-cavities on coverslips were activated with $O_2$ plasma (Diener); (ii) substrates were incubated with 5 μg/ml laminin (Sigma 11243217001) for 1 h at room temperature followed by washing steps; (iii) cells in suspension were centrifuged 3 times at 200 g for 3 min on the samples to direct cells

inside micro-cavities; (iv) the coverslips were then rinsed gently to get rid of the excess of cells between cavities; (v) 15 μl Matrigel (Corning, 356231) was added on top of the sample. After solidification of the Matrigel, the relevant media were added depending on the cyst types. For pancreas spheres, single cells were dissociated from the E13.5 pancreases and immediately seeded in the micro-well without centrifugation.

System-specific media were added to obtain pancreatic spheres or epiblasts. Pancreatic sphere was formed by using DMEM/F12 (ThermoFischer 31331028) with B27 (Gibco 17504-044), recombinant Human FGF2 (R&D 233-FB-025), Y-27632 (Sigma Aldrich ab120129) and Penicillin-Streptomycin (Gibco 15070-063)[44]. Epiblasts were differentiated by using 1:1 ratio of DMEM/F12 (ThermoFischer 31331028) and neurobasal medium (Gibco 21103049) containing 0.5x N2 (Gibco 17502048), 1x B27 with vitamin A (Gibco 12587010), 1x NEAA (Gibco17504044), 0.1 mM 2-Mercaptoethanol (Gibco 21985023), 0.15 mM Sodium Pyruvate and 0.2 mM LGlutamaine (Life Technology GmbH 11360039).

## Immunostaining

For immunostaining, we followed standard protocols Greggio, C. et al. [45,49]. Briefly, samples were washed with PBS and fixed with 4% paraformaldehyde (Electron Microscopy Sciences 15710) diluted in PBS for 15 min. Cells were permeabilized with 0.5% Triton-X-100 for 15 min and then a blocking solution made of 1% Normal Goat Serum in PBS 1X was added overnight. Primary antibodies were added directly to the blocking solution for two days at 4 °C. Following 3 successive washing steps with PBS, the samples were stained with the relevant secondary antibodies for 2 h at room temperature. We used the following primary antibodies: Anti-E-cadherin (Abcam, Ab11512 and Ab53033), Mouse monoclonal Anti-Podocalyxin (BIO-TECHNE SAS, MAB1556-SP), Alexa Fluor Phalloidin 488 (Thermo-Fisher, A12379) for F-actin, and DAPI (Sigma MBD0015) for the nucleus. Samples were washed three times in PBS and mounted on a home-made sample holder system for imaging and conservation.

## Microscopy

In order to track the number of lumens in MDCK cysts, MDCK cells expressing Ecad – Podxl were used to visualize adherens junctions and the apical side, respectively. After cell seeding, images were taken at an interval of 24 h using Leica DMI8 with an Evolve 512 camera coupled to a spinning disk microscope (CSU W1) with a 25x water objective (NA = 0.9) and 63x glycerol objective (NA = 1.2) using the software Metamorph for image acquisition. Positions were chosen based on initial cell numbers, and the same MDCKII cysts were acquired in 3D every 24 h. This was possible with the cavity map reported above. For pancreatic spheres and epiblasts, initial conditions were controlled with the same method and samples were fixed and stained with the relevant antibodies before 3D acquisition with the same microscopy setup.

To record the lumen formation in MDCK, MDCK cysts were imaged at a time interval of 5 min for more than 24 h in a setup regulated for 5% $CO_2$ and 37 °C temperature. We used MDCK cells stably expressing E-cadherin-GFP and podocalyxin as readouts for cells junction and lumen. To record the lumen nucleation in pancreatic spheres, we imaged them live after cells seeding with phase contrast microscopy (Fig. 3d). For the characterization of lumen nucleation in epiblasts, we fixed samples at every 8 h after cells seeding and we stained for E-cadherin and podocalyxin prior to imaging (Fig. 3f).

For lumen fusion events in MDCK cysts, we used a 63x glycerol objective (N.A. = 1.2). Different organoids were selected as starting positions and imaged every 30 min for up to 3 days. Z-stacks (60 μm range, 1 μm step) were acquired. For lumen fusion in epiblasts, epiblasts were imaged from day 2 with SiR-actin (TEBU-BIO, 251SC001) after 1 h incubation before the experiment. Then SiR-actin mix was incubated with media, leading to a final concentration of SiR-actin and Verapamil. The concentrations were larger for MDCK cysts for optimal visualization. Lumen interaction events were recorded with a Leica CSU-W1 spinning disk (63× objective) for more than 10 h at an interval time of 30 min.

## Perturbation experiments

For the inflation experiment[22] (Fig. 7a), MDCK cysts were used after 5 days of culture. We first removed gently Matrigel from the top of micro-wells with a needle. The lumen was inflated by flowing media containing fluorescent dextran (Fisher Scientific SAS D22914) in the relevant media with a micro-pipette and images were taken using Leica spinning disk equipped with micromanipulators. For osmotic shock, we incubated the sample with 300 mM D-mannitol solution (Sigma M9546).To investigate the role of adhesion in lumen fusion, we used 5 mM EDTA and E-cadherin 10 μg/ml antibody[27,28] (Fig. 7b). Snapshots of MDCK cells were taken right before addition of EDTA/E-cadherin antibody, and then the cysts were acquired in 3D for 60 min to record the fusion events. To interfere with cell motion during fusion in

epiblasts or MDCK cyst, we used the myosin inhibitor blebbistatin (Sigma B0560) at 10 μM with a DMSO sample for control. For the hydrostatic pressure measurement and lumen drainage, we used our established method. Hydrostatic pressure was calculated based on the Hagen-Poiseuille model. Channel dimensions were confirmed through combined fluorescence and transmission light imaging, ensuring accurate assessment of channel size and pressure parameters[22,23]. Surface tension measurements were based on established methods[50]. Briefly, borosilicate glass micropipettes were prepared using a pipette puller (Sutter Instrument P2000) and finalized with a Narishige microforge to reach 25–40 μm tips diameter. They were passivated with FCS or PLL-g-PEG and mounted on a micromanipulator connected to a pressure controller (Fluigent Flow EZ −69mbar pressure controller). The full monolayer was tested in the measurement.

## Imaging and data analysis

To extract the shapes and measure the volumes and surfaces of lumens and organoids, we used LimeSeg Fiji Plugin and Skeleton Seg. The segmented 3D structures were saved and visualized. Analysis and quantification were performed with Paraviewer[51].

The number of lumens were rescaled with respect to the cell cycle (Figs. 3a and 4a) by dividing the number of lumens by the duration of each cell cycle. We took 18 h for MDCK, 18 h for pancreatic spheres, and 9 h for epiblasts. We checked that doubling times were consistent with the number of cells counted in each system. In addition, we define lumen as a fluid-filled cavity between cells within the optical resolution, together with the accumulation of apical markers such as podocalyxin and F-actin. Lumen index is defined as the ratio between the area of lumen and the area of the outer shell of the cyst. They were measured for each system. The numerical lumen index was measured by taking the 2D lumen surface over the cyst surface. To compare the fusion process between MDCK cyst and epiblast, we ellipse fitted the outer contour of cysts to extract the long and short axis. Then, we plotted the major axis over the minor axis as a function of time (Fig. 4f, i). To characterize the associated cell motion during fusion, we plotted cell trajectories by using tracking their centers (Fig. 4h, k).

Data were plotted using a written Python code and GraphPad Prism. Statistical tests were performed using non-parametric Mann-Whitney test (two-tailed) to evaluate the significance between lumen formation after cell division and when cells met (Fig. 3c, e), lumen occupancy (Fig. 5b), intensity of E-cadherin (Fig. 5c), and the difference of number of lumens between WT MDCK cyst and ZO-KO cyst (Fig. 6h). The differences of number of lumens before and after EDTA+ anti-E-cadherin treatment were assessed for statistical significance by using Wilcoxon matched-pairs signed rank test (Fig. 7e). Differences between groups for the pipette inflation experiment were analyzed for statistical significance by ANOVA. Statistical significance was indicated using the following symbols: ns $p$-value > 0.05, *$p$ < 0.05, **$p$ < 0.01 and ***$p$ < 0.001.

The number of experimental repeats and a number of cysts or organoids are indicated in the Figure captions. Schematics were generated with Adobe Illustrator.

## Theoretical model and numerical simulations

The cyst was theoretically modeled based on the multi-cellular phase field model with lumen[25], but additionally incorporating the extra-cellular matrix around the cyst. To carefully control the surface tension of each entity (each cell, lumen, and ECM), we applied the resharpening method proposed in refs. 52,53, which enabled us to eliminate the surface tension artificially generated due to the construction principle of the phase field model. The details of this theoretical model are given in the Supplementary Information.

## Statistics & reproducibility

At least 10 cysts or organoids from 3 biological repeats were performed and analyzed.

## Data availability

Source data are provided with this paper. All materials will be available upon reasonable request. Source data are provided with this paper.

## Code availability

The code used for the simulation is available on Github, https://github.com/kana-fuji/MCPFM_for_Lumen_Fusion.git. The code to calculate the free energy for N-cells and lumen is available on GitHub, https://github.com/sano-msk/Free_Energy_Lumen_Ncells.

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

## Acknowledgements

We thank the Riveline group for help and discussions, and Erwan Grandgirard and the Imaging Platform of IGBMC, Patrick Reilly, for assistance in editing. We also thank Byung Ho Lee and Allison Lewis for experimental, conceptual, and image analysis advice and discussions. We thank Benoît Ladoux for providing the MDCKII E-cadherin KO cell line. L.L. and T.G. are supported by HFSP and by the University of Strasbourg and by la Fondation pour la Recherche Médicale. K.F. is supported by an HFSP grant. D.R. acknowledges the Interdisciplinary Thematic Institute IMCBio, part of the ITI 2021-2028 program of the University of Strasbourg, CNRS, and Inserm, which was supported by IdEx Unistra (ANR-10-IDEX-0002), and by SFRI-STRAT'US project (ANR 20-SFRI-0012) and EUR IMCBio (ANR-17-EURE-0023) under the framework of the French Investments for the Future Program. M.S. acknowledges the National Natural Science Foundation in China (12250710131,12174254). T.H. acknowledges the Seed fund of Mechanobiology Institute. A.G.B., A.H., M.S., and DR acknowledge the Research Grant from the Human Frontier Science Program (Ref.-No: RGP0050/2018).

## Author contributions

D.R., A.G.B., A.H., and M.S. supervised the study. L.L and D.R. designed experiments and L.L. performed them with support from T.G., M.L., M.A., Y.A., S.Y., H.P., C.M.L., A.G.B., and A.H. K.F., T.H. M.N., and M.S. designed simulations, and K.F. performed them with support from T.H., M.S., S.T., M.N. D.R., L.L., K.F. and M.S. wrote the manuscript with inputs from all authors.

## Funding

## Competing interests

The University of Strasbourg has filed for patent protection (PCT/EP2025/053648) on the organoid technology used here, and L.L., M.L., and D.R. are named as inventors on those patents. The other authors declare no competing interests.
