## [Transparent Peer Review file · Nature Communications]

Generic comparison of lumen nucleation and fusion in epithelial organoids with and without hydrostatic pressure

Corresponding Author: Professor Daniel Riveline

Version 0:

Reviewer comments:

Reviewer #1

(Remarks to the Author)

The manuscript is well written with sufficient clarity and logical flow. Using a variety of cell types: MDCK (cell line, urinary system, kidney), pancreatic sphere (primary, gastrointestinal, fetal pancreas), mESC (cell line, stem cell), the authors develop a theoretical model to investigate the interplay between pressure, cell-cell adhesion and cell division in driving the distinct fusion dynamics of lumen in these 3 systems, with the hope to uncover the 'generic' rules of lumenogenesis in epithelial organoids. While the aim is noble and the approaches taken are commendable, more work is required to be done (see below) before the manuscript can be considered for publication:

Major:

- The title 'Generic rules ...' sounds rather strong and perhaps an overclaim, since 1) it's not clear if other epithelial organoids (e.g. intestinal organoids) and stem cell aggregates exhibit the same generic features of lumenogenesis shown in this study, and 2) the model has to incorporate additional features and parameters to explain the distinctly different lumen dynamics in the epiblast system, which argues against the generic nature of the model - perhaps something like 'a theoretical framework to understand ...' will be more appropriate?
- Line 137-140: the authors suggest that large Lumen Index (LI) corresponds to larger pressure, but this has not been experimentally measured or tested. Yet this seems to be a crucial assumption for the model simulation. Given that luminal pressure is an important parameter in the model, I would propose the authors to perform some basic pressure measurement method (ablation, micropressure probe, etc). If this not feasible, at least well established methods to perturb pressure and monitor changes in lumen fusion dynamics to see if this fits with the model prediction.
- Line 142-145: I am not sure if it is valid to use E-cad intensity to correlate to adhesion force. Furthermore it is not known whether E-cad is the main contributor to the fusion between two lumens. For example, in Fig 6, high apical actin may suggest that apical constriction or membrane expansion may independently or synergistically influence the lumen fusion dynamics.
- Figure 4a: For fusion rate, a better timescale for normalization related with cellular viscosity or fluid flux is more reasonable. Or just analyze without normalization.
- Figure 4c-e should have more biological repeats with SD indicated.
- Figure 4c: If the fusion process is pressure driven, why does the neighbouring lumen shrink and contribute to the increase of the dominant lumen? Specifically, are the lumen pressure in small and large lumen different?
- Figure 5c: Epiblast simulated timeframe does not appear to capture the experimental results? Can the authors comment on this?
- Figure 5d: On the same point, fusion profile flipped for MDCK and pancreatic spheres in the simulation. Have I missed out anything?

- Figure 6a: The timescale for such inflation is very transient (~1 min), so how would this match the effect of a higher luminal pressure and their impact on more fusion events at longer timescale (days) in the experiments? More bluntly put, does this transient increase in pressure simply rupture local cell-cell junctions at the point of injection? Can the increase in pressure be precisely controlled? This may also explain why there appears to be no shrinkage of either lumens during dextran injection.

- Figure 6a: In line with above comment, I wonder why pressure perturbation other than direct, brutal inflation is not attempted. For example, if as the authors suggest, pressure increase arises from ion-pumping, then one could use small molecule inhibitor such as ouabain to target Na/K ion pump activity, or osmotic perturbations to change the osmotic gradient and monitor the pressure change. In my view, this is more similar to changing the pressure parameter in the model and study the corresponding change in lumen fusion dynamics.

- Figure 6g is missing.

- Figure 6g-h: Why use ZO-KO and not E-cad overexpression? Also in line 202 - why does ZO-KO lead to faster fusion or less nucleation? This is not intuitive and not well explained.

- Line 157-179: There is a general lack of explanation on the essence and assumptions of the model as well as the chosen parameters. And they only comment on the possible extension to other systems of this "framework of numerical simulations" in the last sentence in discussion. For completeness, they should discuss the validity and limits of the model in the discussion section as well. For example, cell mechanics (actomyosin contractility) could potentially play a major role in cellular rearrangement and fusion dynamics, but this seems to be not discussed or considered in the model (or experiments) at all.

- Line 169-170: The authors suggest that the results in Figure 5b and 5d magenta with lower pressure correspond to epiblasts, but their simulation does not recapitulate experimental results in terms of lumen no. with time, lumen index with time, or the lumen structure shown in the simulation snapshots. After specifying the two cyst systems with a local force balance confinement, they show the possibility of this model to reproduce epiblasts features somehow, although it still does not quite capture the time evolution of lumen no. directly. If there exists crucial challenge for them to reproduce epiblasts dynamics in multi-cyst system, they should explain it. Otherwise, a multi-cyst simulation is desired to align with the simulation results for MDCK and pancreatic.

- References missing: epiblast have been shown to form from apical actin polymerization (Dhiraj et al. (2023) bioRxiv). Also, authors should include previous studies on luminal pressure and relevant review (Chan et al Nature <https://doi.org/10.1038/s41586-019-1309-x>, Chan et al. Dev Cell. <https://doi.org/10.1242/dev.181297>)

Minor:

- Line 61-62: "We use epithelial organoids as paradigms for lumen dynamics with physiological relevance^{14,15}". I recommend to remove this sentence or provide new citations. Citations are for intestinal organoids and does not have much physiological relevance to pancreatic duct and epiblast.

- Figure 2a: Typo in figure. "nuclues"

- Fig 4 g, j, the definition of these distance needs better explanations. Alternatively create a simple schematic to show this visually.

Reviewer #2

(Remarks to the Author)

In this manuscript, authors want to clarify generic rules of lumen nucleation and fusion in epithelial organoids by using their unique microfabrication and theory. This is an interesting approach and the obtained results seems to be informative for the researchers in this area. However, due to several problems and its very narrow focus, I cannot recommend its publication in this journal, which is highly influential in a wide range of scientific fields.

The following questions are valuable in updating this manuscript.

Major comments:

1. How to simulate luminal pressure in Fig.5b and confirm this simulation is reasonable by experimental evidence? This is significant point in this manuscript, and thus experimental support of the simulation is necessary.

Additionally, talking about luminal pressure and mechanical cell interactions without actually checking the cell mechanical markers or mechanosensors weaken the study. At least the Rho/ROCK signalling, one of the main pathways mediating the cytoskeletal responses to mechanical signals or integrins should be analyzed. It would have been interesting to follow the spatial and temporal expression pattern of these markers to strengthen the study.

2. The authors focused on only initial cell number to find the generic rules, but I believe that "cell number" is also one of the

important factors. Why they did not count cell number during incubation time even Figure 2a visualized nuclei? Time-dependent cell number change in relation to cell cycle is much important to understand cell luminal pressure.

3. There are lack of quantitative biological analyses such as a cell cycle, cell-cell junction, cytoskeleton, cell marker, and cell adhesion by gene and protein expressions. These biological analyses closely relate to cell proliferation, cell-cell meet, cell growth, and lumen formation.

Additionally, adding more statistical analyses will strengthen the results to support the raised correlations more robustly.

4. Unfortunately, discussion of this manuscript is not enough. They just described only phenomena...I could understand "phenomena", but not understand the generic "rules". Also, consider discussing the implications of these findings for understanding organogenesis and morphogenesis across different developmental contexts to enrich the discussion section.

Minor comments:

1. English grammar should be improved by English native speaker.

2. Fig.1: Please add cell number changing against culture time.

3. Fig.1e: Statistical analysis and error bars are necessary.

4. Fig.1c: Please discuss the reason why all samples even at different initial cell number reached to a "single" lumen.

5. Fig.1: Please show 3D reconstructed image of MDCK spheres. For the 16 initial cells it is not clear whether they really reached the single lumen stage.

6. Fig.2b,d: Please add cell number changing against culture time.

7. Fig.2b,d: Statistical analysis and error bars are necessary.

8. Fig.2: Please show 3D reconstructed image of pancreatic spheres and epiblasts. Especially for the epiblasts whose lumen look quite flat and not really lumen.

9. Fig.2c: It is difficult to find lumen such as the sample of 1 of initial cell number at 48h. What is the threshold or criteria of lumen structure?

10. Fig.3a: There is no description evaluation method of cell cycle. Providing insights into how differences in cell cycle lengths might influence lumen formation dynamics would be beneficial.

11. line 119: How to understand 10 cells in Fig.3f? I can recognize 7 nuclei in DAPI image. It is hard to understand cell number correctly from these images. Thus, 3D reconstructed confocal images are useful to understand cell number and location correctly.

12. Figure 4: Analysis of E-cadherin levels using only immunofluorescence measurement is not enough. PCR or Western Blot should be used to quantify and confirm this hypothesis.

13. Figure 5: Clearly articulate the objectives of the simulations and how they complement experimental findings. The selection of the computational model should be more explained and discussed to understand the rationale behind choosing the phase field model.

14. MDCK II cell line is from dog, while pancreatic spheres cells and embryonic cells both come from mouse, how these cells can also represent the human embryogenesis? This point or possible limitation should be discussed.

Reviewer #3

(Remarks to the Author)

The manuscript by Lu et al. describe similarities and differences when studying lumen formation by three different cell types. These are MDCK cells, E13.5 pancreatic epithelial cells and mouse embryonic stem cells (ESCs). Using these cell types, the authors wish to mimic lumen formation in kidney tubules, pancreatic ducts and blastocysts. They use fluorescently labelled cells, so that lumen formation can be followed dynamically over time. They also follow lumen formation by starting with only one cell per well and ending up with 16 cells per well. Lu et al. conclude that a small lumen first develops between adjacent cells (at least two cells in number) - they term this process "lumen nucleation". Subsequently, several of these small lumens coalesce to form a single large lumen in a process that they term "lumen fusion".

While these findings represent similarities between the three cell systems, they also found differences. Most notably, the cells around the lumen arrange differently: while ESCs form multiple layers around the lumen, the pancreatic cells and MDCK cells only form a single layer (but with the MDCK cells forming more what looks like a columnar epithelial cell layer while the pancreatic cells form a flat epithelial cell layer). Many of the images and image sequences are accompanied by quantification of multiple lumen parameters, and the authors also include simulations of lumen formation processes. Finally, the authors manipulate their cell systems by injecting fluid into 1 of 2 lumens of MDCK cells or blocking cell connections by interfering with tight and adherens junctions. They observe that junctions are an integral part of lumen formation.

In general, the images are stunning, and the system to develop a lumen in a defined setting is straight. Still, the novelty of the presented findings is somewhat limited, given that the formation of a single lumen from small lumens have been described in many publications over the last decades and given that adherens junctions have similarly described as being essential for the lumen formation process. In the opinion of this referee, a substantial amount of work is needed to lift this manuscript with its beautiful images and quantifications/simulation to a stage of innovation that is needed for publication in a top journal.

Major points:

(1) The authors work on cysts rather than tubes, the latter being the prevailing lumenized structures in multicellular organisms. The cyst certainly is relevant for blastocyst formation (or for pathological kidney cysts). In this way, only the ESC work has a strong physiologic counterpart in vivo. Thus, the authors must try to get cell cords as a starting point to develop physiologic kidney tubules and physiologic pancreatic ducts rather than develop kidney cysts and pancreatic cysts in vitro. Given that they work entirely on an in vitro system with no evidence that their findings are relevant for the situation in vivo, they must either develop tubes in vitro and/or study lumen formation in vivo to validate their in vitro findings on cysts.

(2) The role of pressure inside the lumen is speculative. The whole lumen fusion process could be explained by reducing surface tension. As a matter of fact, many tubes (such as blood vessels) are leaky when forming a lumen, meaning that an increased pressure inside the lumen cannot be a generic rule for lumen formation. Here the authors must completely revise their model or at least experimentally work on an alternative model and discuss it.

Minor points:

(1) The authors do not walk the reader through their Figures, but often skip mentioning Figure panels in the main text, which must be changed.

(2) The interpretation that pressure differences may explain the differences between the lumen indices is likely wrong and does not pay any attention to the simple fact that the morphology of the different epithelial cell layers (flat, columnar and multi-layered) can (at least in part) explain these differences.

(3) In Figure 6a, dextran is not visible. No quantification is provided. The authors are advised to transfer the lumen fluid from one MDCK cyst to another, as to avoid that the fluid composition injected (with potential differences in ion concentrations) triggers changes in lumen development.

(4) Given that the authors reveal differences between the cysts they develop in vitro from three different cell types, can they convert MDCK cysts into multi-layered epiblast cysts and vice versa by experimentally manipulating these cysts? At least, this would be more innovative than just showing that junctions are needed for proper lumen formation.

(5) The authors speculate about an ion pumping mechanism involved in coalescence of small lumens. Which ion pump do they think of? Again, this referee is not convinced about pressure differences driving major changes in the developing lumen. If they insist though, they must use KO cells for a specific ion pump and provide evidence for its role in lumen formation in their system.

Reviewer #4

(Remarks to the Author)

This manuscript from Lu and colleagues proposes generic rules of lumen nucleation and fusion in epithelial organoids. The number of lumens, and lumen fusion, are observed in MDCK cells, pancreatic cells, and mouse derived embryonic stem cells as a function of cell number and culture time, with the observation of multiple nucleation events and an initial increase in lumen number in larger cell clusters followed by cell fusion. In MDCK and Pancreatic clusters, lumens form during cell division and cell meeting events, whereas lumen nucleation occurs in rosettes for the mESCs. Lumen fusion in MDCKs and pancreatic spheres is associated with increase in lumen index, whereas lumen formation in the mESCs is associated with cell rearrangements. Simulations are performed of nucleation and fusion based events for the three systems. Inflation via a pipette promotes fusion events in MDCKs, and decreased cell-cell adhesion (via molecular agents or knockout) promotes fewer lumens in MDCKs. The authors conclude that fusion dominated by increase in pressure in pancreatic and MDCK spheres whereas epiblast lumens fuse by cell motion.

The study of lumen fusion is quite interesting, and there are some nice elements to this study, in particular the control of starting cell number and the detailed monitoring of fusion. However, there are a number of major concerns that dampen enthusiasm for the manuscript and its suitability for Nature Communications in its current form. Mainly, there are limited mechanistic insights resulting from this largely observational study, and some of the conclusions are currently not supported.

Comments:

1. Physiological relevance: One point of confusion is the relevance of these lumen fusion events for biology. One would expect likely the 1 cell case to be most relevant in this regard. This should be explained so the reader can better understand the physiological relevance of the observations.

2. Mechanistic insights: The studies are largely observational, observing lumen nucleation and fusion, with only the inflation studies and e-cadherin perturbation studies pointing towards some mechanistic insights. I do not believe the authors have supported the conclusion that "lumens fuse by an increase in lumen volume for pancreatic spheres and MDCK cysts, whereas cell convergent directional motion leads to lumens fusion in epiblasts."

a. Just because artificial inflation promotes fusion doesn't necessarily mean that pressure is driving fusion in the natural structures. For example, I would anticipate that inflation would drive lumen fusion in the mESC as well. That study should be performed. Further, individual lumen size should be tracked prior to fusion (this may be related to lumen index but this is not clear). Finally, natural lumen expansion should be inhibited (e.g. via increased osmotic pressure) to further support this conclusion.

b. That cell-cell junctions are a barrier to lumen fusion would not be surprising. However, the data in Fig. 6e appear to show that these various perturbation reduce initial lumen nucleation and not necessarily lumen fusion. This needs to be analyzed

and clarified.

c. The conclusion on convergent directional motion leading to lumen fusion in “epiblasts” is based on correlative/observational data, and it is not clear what the driving factor is. For example, it could be that cell rearrangements are passive and the consequence of pressure-driven fusion events. To show a causal relationship, the cell movement would need to be modulated and the impact on fusion assessed.

d. At a fundamental level, it is unclear what is driving the hypothesized pressurization and convergent directional motion. The authors could examine the role of osmotic pressure for example (image ions, perturb ion channels, and increase osmotic pressure in media), and the role of contractility (imaging of the contractile actomyosin network, and inhibition of its activity).

Substantial mechanistic studies are needed to clarify the molecular drivers of fusion in these different contexts. Further, cell division could be perturbed (i.e. with an agent like mitomycin C) and the impact on cell division and fusion events monitored.

e. Related to the pressurization question, it was not obvious to this reviewer what the significance of the LI parameter was. I am wondering whether solidity would be a better measure of the role of pressure, as described in a manuscript by Vasquez, et al., Nature communications 2021 (“Physical basis for the determination of lumen shape in a simple epithelium”). Further, lumen area alone could be shown.

3. It was difficult to follow the model and what it adds to the paper. If lumen nucleation and volume, and each of the observable parameters is part of the input assumptions, it is not surprising that the authors capture behavior of the different systems, especially with a different model used for the epiblasts. What is the specific question that the model is trying to address, and how is it validated? What are the input assumptions, how are the parameters estimated, and what are the constraints? This section should be expanded/clarified and these questions addressed.

Minor comments:

-The term epiblast is likely inappropriate. At best, this is a model of the epiblast, though it is not clear how good of a model. This should be clarified and the use of some qualifiers should be included.

-Fig 3a – difficult to understand what are the data and what is the fit. Also, the R2 value, or other appropriate statistic for fit should be included).

-Fig. 3g – what about the multi-lumen clusters?

-Fig. 4c – is this averaged data? Are there statistics here.

-6b/c – do lumens also leak out into the surrounding extracellular space in this experiment?

-Inflation studies require some kind of quantification ideally and statistical tests to make stronger conclusions.

Reviewer #5

(Remarks to the Author)

In this article, Linjie Lu and colleagues aim to identify the common dynamics in lumenogenesis across three different systems: MDCK cysts, pancreatic spheres, and epiblast cysts. This unifying observation is achieved through the integration of quantitative cell biology, microfabrication techniques, and theoretical frameworks. In this sense, the authors firstly report that the maximum number of lumens is dictated by the initial cell count, without influencing the steady state, resulting in a final single lumen. They then describe two phases: a nucleation phase followed by a fusion phase. Lumens emerge between two cells in pancreatic and MDCK cysts during the nucleation phase, while in epiblasts, they form at the rosette stage between ten cells. During the subsequent phase, lumens in pancreatic spheres and MDCK cysts fuse through an increase in lumen volume. In contrast, in epiblasts, the fusion results from the convergent directional motion of cells. To support these findings, the researchers replicated the dynamics of lumens numerically using a phase field model that incorporates basic rules for cell proliferation, cell adhesion, and lumen growth. By manipulating cell adhesion and lumen volume in MDCK cysts, the authors were able to successfully replicate the fusion dynamics observed in pancreatic spheres and epiblasts.

This study employs a comprehensive approach by investigating lumen dynamics across multiple cellular systems (MDCK cells, pancreatic spheres, and mESC-derived epiblast organoids). This breadth allows for a comparative analysis, enhancing our understanding of common principles underlying organogenesis. By combining experimental techniques such as cell tracking with numerical simulations, the study provides a robust framework for studying lumen formation. This integration allows for the validation of experimental findings and the exploration of underlying mechanisms in a controlled computational environment. Indeed, we are presented with a groundbreaking mathematical simulation that appears to integrate the three models discussed in the article. Also, The study includes experimental perturbations to validate the observed mechanisms of lumen formation. By manipulating factors such as lumen growth rate and cell-cell adhesion, the researchers provide empirical evidence supporting their computational and theoretical predictions.

Overall, the study's strengths lie in its comprehensive approach, integration of experimental and computational methods, quantitative analysis, identification of both differences and commonalities across cellular systems, experimental validation of findings, and potential implications for tissue engineering. These strengths contribute to a deeper understanding of lumen dynamics in organogenesis and pave the way for future research in the field.

However, While the article presents a comprehensive study on the dynamics of lumen formation in organoids, there is limited new information or groundbreaking discoveries presented in the research. The study largely confirms existing knowledge and observations regarding organogenesis and lumen formation. It acknowledges previous research on the subject and primarily seeks to validate and compare these findings across different cellular systems. Indeed, Many of the mechanisms and processes described in the study, such as the role of cell proliferation, cell-cell adhesion, and luminal pressure in lumen formation, are well-documented in existing literature. The study does not introduce significantly novel mechanisms or pathways involved in organogenesis. Hence, given that it is widely recognized that various facets of lumen

formation are common across these models, the presence of a biphasic behavior is not surprising (2). This commonality also extends to the acknowledged roles of cellular junction remodeling and luminal ion pumping.

In addition, aside from the mathematical simulation, the authors introduce a compelling point by highlighting that in pancreatic spheres, the lumen index surpasses that of MDCK cysts, and fusion is likely driven by luminal ion pumping. In contrast, MDCK relies on cell-adhesion forces, while epiblast fusions are primarily governed by cellular motion. To improve clarity, it would be beneficial for the authors to explicitly explain each dominating mechanism for each model throughout the entire text, instead of only in the discussion section.

In sum, while the study may provide some additional insights into the specific dynamics of lumen formation in different cellular systems, the findings are largely incremental rather than transformative. The observed differences in timing and mechanisms of lumen formation across systems, while interesting, may not represent a significant departure from existing knowledge.

Minor points

- 1.) Fig. 4B: Sir-Actin is used as staining for epiblast cysts but phase contrast for pancreatic spheres. To improve image quality and clarity, I would suggest the use of Sir-Actin in pancreatic spheres too.
- 2.) General review of English language is advised.

References:

1.) Carleton AE, Duncan MC, Taniguchi K. Human epiblast lumenogenesis: From a cell aggregate to a luminal cyst. *Semin Cell Dev Biol.* 2022 Nov;131:117-123. doi: 10.1016/j.semcdb.2022.05.009. Epub 2022 May 27. PMID: 35637065; PMCID: PMC9529837.

Version 1:

Reviewer comments:

Reviewer #1

(Remarks to the Author)

The authors have managed to address our queries in the rebuttal and they have done a lot more experiments to support their claims now. They also changed to a more appropriate title and in terms of the work done so far, it is quite complete for their claims.

The irony is that the title also accurately reflected how narrow the work is. Probably only the motion-directed lumen fusion in epiblast is relatively novel.

The pressure measurements appear sound, although a value of close to 0 Pa for the case of epiblasts may suggest that the approach to measure pressure may not be sensitive to measure very low pressure. The tension measurements are not adequately described in the main texts nor the methods. Also, in the revised manuscript there are still mistakes like missing mention of sub figures in figure legends and referencing the wrong figures in the main text.

Reviewer #2

(Remarks to the Author)

The authors addressed nearly all technical and conceptual concerns raised in the initial review, providing additional experiments and data analysis to validate their claims and a more detailed discussion of the results' implications.

However, there is still a lack of mechanistic depth and an overreliance on observational data. Here are the aspects that still need improvements:

- About the simulation of luminal pressure: The authors' explanation is reasonable. However, the connection between ξ and pressure could be more explicitly tied to the observed data. It would be better to add clarification on whether the model's assumptions (e.g., constant osmotic pressure) align with biological variability.

Also, please show how hydrostatic pressure differences actually correlate with lumen size and shape, to validate the model as following the Laplace law. Same can be done for hyperosmotic or hypoosmotic solutions (e.g., adding D-mannitol or sucrose to the medium to perturb osmotic gradients and monitor changes in lumen size, fusion rates, and dynamics).

Alternatively, the authors can also use inhibitors of ion channels or pumps to disrupt osmotic pressure regulation and again observe the effects on lumen growth and fusion.

- Impact of cell number on luminal pressure and dynamics: The cell number was measured indirectly using the lumen index and data correlate only initial cell numbers with lumen dynamics. While the authors' explanation seems logical, it still lacks

direct evidence correlating cell number changes during culture with pressure. A more detailed time-course data on cell proliferation or more justification for their reliance on indirect measures is needed for this study. At least the final real cell number should be provided.

- Lack of biological analysis: It is understandable that future studies will integrate these biological factors. However, the current study relies mostly on images analysis and more quantitative data are needed to enhance the robustness of the conclusions by mechanistic studies. The lack of molecular and biological analyses currently limits the study's broader biological relevance and the relevance for a publication in this journal.

The authors should add immunostaining for mechanosensitive markers (e.g., YAP/TAZ nuclear localization, Rho/ROCK signaling, actomyosin organization) as well as RNA or protein expression analysis of aquaporins or ion transporters involved in maintaining osmotic gradients. Alternatively, inhibiting actomyosin contractility with drugs like blebbistatin or Y-27632 (ROCK inhibitor) can be done too.

Reviewer #3

(Remarks to the Author)

The authors have revised their manuscript by including a number of new experiments, including a comparison of hydrostatic pressures across different cell systems and micropipette inflation experiments. Figure L16 shows an elegant way to create lumens in cell cords, and this Figure must be included in the final manuscript (and not just shown for the referees). The surface tension measurement, however, is not addressing ref. 3 concern that a reduction of luminal surface tension is a key driver, as the authors measure surface tension at the basolateral rather than luminal, apical cell surface. If the authors include L16 in the published paper, the manuscript is fine.

Reviewer #4

(Remarks to the Author)

In the revised manuscript, the authors have been partially responsive to my concerns. My sense is that the manuscript is still a bit descriptive and less mechanistic; however, there are lot of interesting experiments and analyses here, and the study of lumen dynamics, particularly fusion, is novel and will be of interest to the field. So, I generally support publication. However, the following points should be addressed.

1. As mentioned in previous review, the term epiblast is not appropriate here – it makes it seem as if live epiblasts are being studied. The references cited from 2014 and 2017 supporting use of this term do not reflect the current standard and terminology of the community. Perhaps epiblast model, epiblast organoid, or synthetic epiblast would be more appropriate.
2. The authors make the claim that the cell convergent dynamics in the epiblast model leads to fusion, and have supported this claim with the blebbistatin perturbation (Fig. 8). I don't think this is the strongest experiment, given that adding blebbistatin is a blunt perturbation, but its ok for this experiment. However, the authors should support this at least with quantification of the experimental phenomena (centripetal movement and fusion rate) rather than simply show images.
3. The origins of lumen nucleation in epiblasts (lines 184 – 190) were not clear to me. Why are the authors arguing that lower lumen pressure leads to this alternate form of lumen nucleation? Is this simply a matter of cell division growth rate and geometrically fitting the growing cells within a spherical structure? The proposed mechanistic insight should be explained clearly here.
4. The abstract notes that "We finally use MDCK cysts to manipulate cell adhesion and lumen volume and we successfully reproduce the fusion dynamics of pancreatic spheres and epiblasts". It is not clear to me where the MDCK cysts are manipulated to induce the centripetal motion to induce fusion, to model the fusion dynamics of the epiblast model. This should be clarified/quantified, or the claim should be removed.
5. At a broader level and related to 4, it still seems to me that the epiblast model behavior is fundamentally different than the behavior of the pancreatic sphere and MDCK cyst (as is also suggested by their final takeaway figure 9). The nucleation behavior, pressurization of the lumen, and lumen fusion dynamics are all quite different, yet the authors are trying to put these different behaviors into a universal framework. I don't think this is really supported by the data, and I don't think it takes away from the paper if they discover two modes of lumen nucleation and fusion. However, I would give the authors some latitude here in what they conclude, but perhaps they could more clearly explain the basis for their conclusion of universality in the discussion.

Version 2:

Reviewer comments:

Reviewer #2

(Remarks to the Author)

The authors have thoroughly addressed all my previous comments and concerns, leading to significant improvements in the clarity, rigor, and overall quality of the manuscript and including very beautiful and convincing figures! The current revisions enhance the scientific soundness and readability of the work, making it suitable for publication in its current form.

Reviewer #4

(Remarks to the Author)

The authors have generally addressed my comments. I am supportive of publication.

REVIEWER COMMENTS

Reviewer #1 (Remarks to the Author):

The manuscript is well written with sufficient clarity and logical flow. Using a variety of cell types: MDCK (cell line, urinary system, kidney), pancreatic sphere (primary, gastrointestinal, fetal pancreas), mESC (cell line, stem cell), the authors develop a theoretical model to investigate the interplay between pressure, cell-cell adhesion and cell division in driving the distinct fusion dynamics of lumen in these 3 systems, with the hope to uncover the ‘generic’ rules for lumenogenesis in epithelial organoids.

We thank the reviewer for the positive evaluation.

While the aim is noble and the approaches taken are commendable, more work is required to be done (see below) before the manuscript can be considered for publication:

Major:

- The title ‘Generic rules ... ‘ sounds rather strong and perhaps an overclaim, since 1) it’s not clear if other epithelial organoids (e.g. intestinal organoids) and stem cell aggregates exhibit the same generic features of lumenogenesis shown in this study, and 2) the model has to incorporate additional features and parameters to explain the distinctly different lumen dynamics in the epiblast system, which argues against the generic nature of the model - perhaps something like ‘a theoretical framework to understand ... ‘ will be more appropriate?

We agree with the reviewer. The claim is that other organoids such as intestinal organoids may exhibit similar rules compared with pancreatic spheres/MDCK cysts. The model now incorporates new features for the epiblast case.

Accordingly, we changed the title “Generic rules of lumen nucleation and fusion in epithelial organoids” for “Generic comparison of lumen nucleation and fusion in epithelial organoids with and without hydrostatic pressure”.

- Line 137-140: the authors suggest that large Lumen Index (LI) corresponds to larger pressure, but this has not been experimentally measured or tested. Yet this seems to be a crucial assumption for the model simulation. Given that luminal pressure is an important parameter in the model, I would propose the authors to perform some basic pressure measurement method (ablation, micropressure probe, etc). If this not feasible, at least well-established methods to perturb pressure and monitor changes in lumen fusion dynamics to see if this fits with the model prediction.

We tested this hypothesis by drainage experiments (based on the method presented in our article Mukenhirn et al, Developmental Cell, 2024). We show below and we report in the new version of the paper the relation between lumen index and pressure for MDCK cysts. We report a similar relation for pancreatic spheres in the following preprint Lee et al, BioRxiv, 2024.

Figure L1: High lumen index is associated with large hydrostatic pressure in MDCK spheres. (a) Drainage is triggered by a local laser cut with time-lapse images showing the midplane of cysts before and after laser cutting. The positions of the cuts are marked by red lines. Two examples are displayed, with time indicated in seconds (s), in green E-cadherin, in red podocalyxin. Scale bars: 10 μm . (b) Relation between hydrostatic pressure and lumen occupancy.

- Line 142-145: I am not sure if it is valid to use E-cad intensity to correlate to adhesion force. Furthermore it is not known whether E-cad is the main contributor to the fusion between two lumens. For example, in Fig 6, high apical actin may suggest that apical constriction or membrane expansion may independently or synergistically influence the lumen fusion dynamics.

To address this question of adhesion strength, we conducted additional experiments and analyses.

First, we performed Western blots to quantify E-cadherin levels on MDCK wild type cells, MDCK overexpressing E-cadherins (Fig. L2a,b) and epiblasts (see below Fig. L2c). Our results show that MDCK cysts with overexpression of E-cadherin exhibit higher E-cadherin levels. They also support that MDCK wild type cells have higher levels of E-cadherin compared to epiblasts. This further supports the notion that adhesion is larger in MDCK cysts compared to epiblasts.

In addition, in order to test adhesion strength, we performed injection experiments on day 2 MDCK cysts and epiblasts (Fig. L2d). These experiments show that a large increase in lumen volume is sustained for MDCK cysts. In contrast epiblasts failed to maintain such a volume change, disrupting upon injection. This result further supports that MDCK cysts possess higher adhesion forces compared to epiblasts.

Finally, we also tracked fusion in MDCK E-cadherin knockout (E-cadherin-KO) cysts to test the effect of weaker adhesion on fusion. These cysts fused faster than wild-type

(WT) cysts (Fig. 7e-g). This supports the result that lower adhesion is associated to facilitated fusion.

While we acknowledge that other factors, such as apical actin constriction and membrane expansion may also play a role, our data consistently show that E-cadherin is a central contributor.

Taken together, our results suggest that E-cadherin is a key factor in adhesion force, contributing to the dynamics of lumen fusion. This is included in the new version of MS.

Figure L2: Adhesion is different between epithelial cysts. (a) Immuno-staining of wild type MDCK and MDCK cells overexpressing E-cadherin, scale bar 20 μ m. Cysts are stained by F-actin(magenta) and nucleus in cyan. (b) Western blots for E-cadherin comparing wild type MDCK cells, MDCK cells overexpressing E-cadherins, and (c) epiblasts. (d) Micropipette inflation experiments on MDCK cysts (left) and epiblasts (right) (see Mukenhirn et al, Developmental Cell, 2024).

- Figure 4a: For fusion rate, a better timescale for normalization related with cellular viscosity or fluid flux is more reasonable. Or just analyze without normalization.

We understand the suggestion of the reviewer. Cellular viscosity or fluid flux could give physical parameters that may allow to compare systems based on the cysts mechanical properties. However, the determination of these parameters is difficult.

To address the concern of the reviewer, we have now included the data without renormalization (Fig. L3a). We also show the data after renormalization (Fig. L3b). We explain the reason in the text: lumens are generated after each division and we take this into account by renormalizing.

Figure L3: Evolution of the lumen number (a) as a function of time, (b) as a function of the number of cell cycle.

- Figure 4c-e should have more biological repeats with SD indicated.

We have now added and quoted more individual examples to illustrate lumen dynamics and lumen fusion for the three systems (see Ext. Fig. 4). We also added more biological repeats (Fig. L4). In addition, we compared 3 systems (Fig. L4), the dark line corresponds to the mean value of the lumen index, and the shaded region indicates the standard deviation.

Figure L4: Lumen fusion dynamics across different systems. (a) Representative examples of lumen fusion in MDCK cysts, pancreatic spheres, and epiblasts, along with quantification of the lumen index. Lumens with low and high lumen index are shown in blue and black, respectively, while the sum is indicated in red. MDCK cysts are labeled with E-cadherin (green) and Podocalyxin (red), while pancreatic spheres and epiblasts are visualized using SiR-actin (gray). Additional examples of lumen fusion are also shown. (b) Quantification of lumen index in another set of examples across the three systems. (c) Mean lumen index for each system, represented by the dark line, with the shaded region indicating the standard deviation (N=3, n=3). Scale bar: 10 μ m.

- Figure 4c: If the fusion process is pressure driven, why does the neighbouring lumen shrink and contribute to the increase of the dominant lumen? Specifically, are the lumen pressure in small and large lumen different?

To address this question, we estimated the pressure in small and in large lumens in cysts containing two lumens (Fig. L5). Our measurements show that pressure in small lumen is larger than in large lumen. This suggests that lumen fusion in MDCK cysts and pancreatic spheres is primarily driven by differential lumen pressures. This is consistent with the increase of the dominant lumen. Theoretically, this is also consistent with the Young-Laplace's law, which states that the smaller the radius, the larger the pressure difference. These new data are included in the new version of MS (Ext. Fig. 6). This does not go against what could be inferred by the analysis of the new measurements of hydrostatic pressure presented in Fig. L1, where the smaller the lumen occupancy the smaller the hydrostatic pressure. Tissue surface tension should be higher for the tissue surrounding small lumens in the two-lumen case than in the single lumen case. Indeed, cells are more stretched in the two-lumen system.

Figure L5: Pressure differences in MDCK Cysts with two lumens. (a) Laser-induced drainage experiment showing time-lapse images of the midplane of cysts before and after laser cutting (red bar). The cut positions are indicated by red lines. The volume of each lumen is tracked over time. (b) Changes in lumen volume over time, with different colors representing individual cysts. Blue circles and black squares correspond to smaller and larger lumens, respectively. (c) Hydrostatic pressure differences associated with each experiment, with the color scheme matching that used in panel (b) for consistency.

- Figure 5c: Epiblast simulated timeframe does not appear to capture the experimental results? Can the authors comment on this?

Thank you for this comment. We recognize that the former simulation parameters did not fully capture the experimental results for epiblasts.

In our study, we observed that both MDCK cysts and pancreatic spheres require a minimum of two cells to create a lumen, whereas epiblasts require 10 cells to form a lumen. Our simulation model was initially set up with the rule that a lumen is created between two cells, which accurately represents MDCK cysts and pancreatic spheres but not epiblasts.

To address this discrepancy, we now treat epiblast in a different way. We measured hydrostatic pressure in the lumen of epiblasts (see Fig. L15). We show that the lumens of epiblasts have a negligible hydrostatic pressure. This changes the rule of nucleation. In this context, we now show that epiblasts require a minimum of 10 cells to generate a lumen (see Fig. L6). To prove this, we investigated the stability of the free energy of the spherical cysts with a single lumen as described in the Supplementary Information (theory). Stability is determined by the competition between the work done by the lumen growth by the lumen pressure and the minimization of surface energy of cell membrane, elasticity of cell and ECM as well. As shown in Fig. L6a, when the pressure is low, lumen cannot grow and zero radius state of the lumen ($r=0$) is stable for $N=2,4,8$ cell cases, while $N=12$ cell state becomes unstable for lumen nucleation as shown by the minimum of the free energy at finite lumen radius. We consider that epiblast corresponds to this case as validated in the experiment by the large leak in epiblast. When the lumen pressure is high (Fig. L6c), $N=2$ cell state becomes unstable for lumen nucleation. We suspect this situation corresponds to MDCK and pancreas cysts. In that sense, three systems obey the same principle that lumen nucleation and growth are determined by the balance between lumen pressure and energy minimization. Our numerical phase field model also evolves to minimize the same type of energy which includes surface tension, elasticity of cell and ECM, and the work exerted by the lumen growth. Specificity of epiblast case lies in its low luminal pressure. The analytical calculation of the hypothesized free energy here is assuming a rosette structure with apical surfaces of all cells meeting at the center. This rosette structure becomes unstable for the lumen nucleation when the number of cells is large enough, even at low pressure. It suggests that cell motion toward the center in epiblast contributes to the

formation of rosette structure before lumen formation. These elements are now reported throughout the MS and in particular in the new version of the Supp. Note.

Figure L6: Free energy $F(r, R_{\min})$: (a) $\Delta p = 0$ [Pa] for $N=2, 4, 8, 12$. (b) $\Delta p = 30$ [Pa] for $N=2, 4, 8, 12$. (c) $\Delta p = 100$ [Pa] for $N=2, 4, 8, 12$. Numerical parameters are $\gamma_A = 100$ [Pa $\cdot \mu\text{m}$], $\gamma_L = 100$ [Pa $\cdot \mu\text{m}$], $R_0 = 20$ [μm], $L = 40$ [μm], $L_0 = 38.5$ [μm], $k = 0.01$, $k_E = 0.02$ for all cases. For $N=12$ case, we assumed regular dodecahedron shape for calculation.

- Figure 5d: On the same point, fusion profile flipped for MDCK and pancreatic spheres in the simulation. Have I missed out anything?

We confirm the fusion profile resulting from the numerical simulations. There was no flipping of the results between MDCK and pancreas spheres simulation results. It looks like so, because apparent lumen occupancy for MDCK ($\xi=0.18$) is slightly larger than that of pancreatic sphere ($\xi=0.20$) in the simulation results shown in Figure 5d (now Fig. 6d), but the time stamps are different as shown in Figure 5b (now Fig. 6b). In Figure 5b (now Fig. 6b), if we compare a similar time point of $t=1700$ for MDCK ($\xi=0.18$) and $t=1600$ for pancreas ($\xi=0.20$), pancreatic sphere has much larger lumens compared to those in MDCK ($\xi=0.18$). If we focus on the timing at which the last two lumens fuse in the two systems, it occurs much later in MDCK ($\xi=0.18$) at around $t=6200$ compared to around $t=2400$ in the pancreas ($\xi=0.20$) where the sizes of the spheres in two systems are very different. Therefore, when looking at the same time, the larger the value of ξ , the larger the size of the lumen and lumen occupancy.

- Figure 6a: The timescale for such inflation is very transient (~ 1 min), so how would this match the effect of a higher luminal pressure and their impact on more fusion events at longer timescale (days) in the experiments? More bluntly put, does this transient increase in pressure simply rupture local cell-cell junctions at the point of injection? Can the increase in pressure be precisely controlled? This may also explain why there appears to be no shrinkage of either lumens during dextran injection.

We designed this approach in parallel and its details are reported in Developmental Cell. The rupture of cell-cell junctions does occur at the point of injection.

Regarding the absence of lumen shrinkage during Dextran injection, it is important to note that images in the submitted version were not included immediately after the

removal of the pipette. To show that lumens shrink, we have included additional data showing the lumen dynamics after the pipette removal (see Fig. L7). We extrapolate to longer timescale to show that breakage of the cell-cell contacts between lumens can be separated by this approach.

Figure L7: inflation experiments. Snapshot of MDCK cysts before injection, after injection, and after pipette removing (about 10s timescale between images).

- Figure 6a: In line with above comment, I wonder why pressure perturbation other than direct, brutal inflation is not attempted. For example, if as the authors suggest, pressure increase arises from ion-pumping, then one could use small molecule inhibitor such as ouabain to target Na/K ion pump activity, or osmotic perturbations to change the osmotic gradient and monitor the pressure change. In my view, this is more similar to changing the pressure parameter in the model and study the corresponding change in lumen fusion dynamics.

We tested these approaches but they do not work with MDCK cells. Instead, we tested osmotic perturbation with osmotic shock on lumens using 300 mM D-Mannitol (Knight H et al, 1998) on day 1 MDCK cysts, each starting from 8 initial cells (Fig. L8). We obtained lumens with multiple lumens at day 1. We then added Mannitol. Our results show that osmotic shock led to single lumen formation, whereas the control configuration typically resulted in multiple lumens at day 3, with an average of 3 lumens per cyst. They are reported in the new version of MS (Fig. 7bc).

Figure L8: Effects of osmotic shock on MDCK cysts with multiple Lumens. (a) Experimental setup (top left): MDCK cysts were cultured for 1 day before being treated under control conditions or with D-mannitol. The culture duration is indicated in days. Representative images of cysts are shown at two time points: day 1 (top right) and day 3 (bottom left) for each condition. Cysts are stained for F-actin (magenta) to visualize structural changes. Scale bar: 10 μ m. (b) Statistical analysis based on three independent experiments ($N=3$), with 79 cysts in the control group ($n=79$) and 198 cysts in the D-mannitol-treated group ($n=198$). $p < 0.0001$.

- Figure 6g is missing.

We corrected in the new version.

- Figure 6g-h: Why use ZO-KO and not E-cad overexpression? Also in line 202 - why does ZO-KO lead to faster fusion or less nucleation? This is not intuitive and not well explained.

Along the suggestions of the reviewer, we conducted E-cadherin overexpression experiments. Compared to wild-type (WT) MDCK cells at day 8, most cysts maintained a multiple lumen configuration rather than the single lumen observed in WT MDCK cysts (Fig. L9).

In addition, we used ZO-KO cysts because both ZO-KO cysts and epiblast share similar morphology with a low lumen index and similar phenotypes. To further substantiate the comparison with epiblasts, we generated a configuration similar to the epiblast (see Fig. L9). We observed directional cell motion and lumen fusion, supporting our hypothesis about the mechanical properties and behaviors of ZO-KO cysts are similar to epiblasts.

We detail more these results in the new version of the article.

Figure L9 Lumen fusion is compared across MDCK wild-type (WT) cysts, MDCK ZO-KO cysts, and epiblasts. The ZO-KO mutants display fusion behavior similar to that observed in epiblasts. MDCK WT and ZO-KO cysts are labeled with Podocalyxin (red) and E-cadherin (green), while epiblasts are stained with SiR-actin (gray). Scale bar: 10 μm .

- Line 157-179: There is a general lack of explanation on the essence and assumptions of the model as well as the chosen parameters. And they only comment on the possible extension to other systems of this "framework of numerical simulations" in the last sentence in discussion. For completeness, they should discuss the validity and limits of the model in the discussion section as well. For example, cell mechanics (actomyosin contractility) could potentially play a major role in cellular rearrangement and fusion dynamics, but this seems to be not discussed or considered in the model (or experiments) at all.

In the new version, we explain in more detail the assumptions of the model, the role of osmotic pressure, hydrostatic pressure, cell surface tension, adhesion, and the chosen parameters in the Supplementary Information, and we also refer to the validity and limits of the model in our associated articles (Tanida et al. BioRxiv, Kana et al. 2022).

Regarding the chosen parameters, our aim was to test the rules of lumen dynamics observed in experiments. Therefore, we focused on tuning key physical parameters affecting lumen dynamics, such as lumen growth driving term (ξ), cell division, and cell adhesion. Time of cell cycle was adjusted to times of each experimental system. Pressure was also derived from measurements. Adhesion strength was also evaluated. With these measurements and estimates, we manage to recapitulate the main experimental features. While we concentrated on the essential physical rules, we acknowledge that contractility

and other mechanical properties could be included in the model. For instance, active cell motion observed in the fusion process in epiblast might be produced by acto-myosin dynamics. We introduced active force to the center of each cell in the order to reproduce cell motion and fusion dynamics in epiblast. Future work should incorporate these factors from microscopic level to provide a more comprehensive understanding of cellular rearrangement and fusion dynamics.

- Line 169-170: *The authors suggest that the results in Figure 5b and 5d magenta with lower pressure correspond to epiblasts, but their simulation does not recapitulate experimental results in terms of lumen no. with time, lumen index with time, or the lumen structure shown in the simulation snapshots. After specifying the two cyst systems with a local force balance confinement, they show the possibility of this model to reproduce epiblasts features somehow, although it still does not quite capture the time evolution of lumen no. directly. If there exists crucial challenge for them to reproduce epiblasts dynamics in multi-cyst system, they should explain it. Otherwise, a multi-cyst simulation is desired to align with the simulation results for MDCK and pancreatic.*

Along our answer above, we now treat differently epiblasts and MDCK/pancreatic spheres, please see our associated modifications in the revised manuscript (and Fig. 5d-g, Fig. 6e-h).

- *References missing: epiblast have been shown to form from apical actin polymerization (Dhiraj et al. (2023) bioRxiv). Also, authors should include pervious studies on luminal pressure and relevant review (Chan et al Nature <https://doi.org/10.1038/s41586-019-1309-x>, Chan et al. Dev Cell. <https://doi.org/10.1242/dev.181297>)*

We now quote these references.

Minor:

- Line 61-62: *“We use epithelial organoids as paradigms for lumen dynamics with physiological relevance^{14,15}”. I recommend to remove this sentence or provide new citations. Citations are for intestinal organoids and does not have much physiological relevance to pancreatic duct and epiblast.*

We modified the references.

- Figure 2a: Typo in figure. “nuclues”

We corrected.

- Fig 4 g, j, the definition of these distance needs better explanations. Alternatively

create a simple schematic to show this visually.

We changed the new scheme to illustrate the definition of the distance.

Reviewer #2 (Remarks to the Author):

In this manuscript, authors want to clarify generic rules of lumen nucleation and fusion in epithelial organoids by using their unique microfabrication and theory. This is an interesting approach and the obtained results seems to be informative for the researchers in this area. However, due to several problems and its very narrow focus, I cannot recommend its publication in this journal, which is highly influential in a wide range of scientific fields.

The following questions are valuable in updating this manuscript.

We thank the reviewer for the comments and we changed the manuscript along the comments.

Major comments:

1. How to simulate luminal pressure in Fig.5b and confirm this simulation is reasonable by experimental evidence? This is significant point in this manuscript, and thus experimental support of the simulation is necessary.

We provide explanation of the parameter ξ and its connection to the measured hydrostatic pressure. In the Supplementary Information, we have analytically shown that the parameter ξ in the phase field model corresponds to the osmotic pressure of the lumen. This is the driving force for the lumen growth. Hydrostatic pressure difference between the lumen and outside the cyst is determined by the force balance between luminal pressure and the tension of cell layer, which takes the form of the Young-Laplace law.

There must be additional effects of ion pumping and related feedback in the lumen growth dynamics. The whole process is governed by many factors in principle. For the sake of simplicity, we assumed that the driving force (osmotic pressure) is constant during the lumen growth, and expansion or shrinkage of the lumen is spontaneously determined by the force balance between luminal variable and tension of cell layer which is automatically attained in the dynamics of the phase field model. That constitutes the main assumption of lumen growth process in our model. This is consistent with the main framework of lumen growth dynamics known in literature (Torres-Sánchez et al. [11]) which is summarized in SI.

During the lumen fusion in the experiment, we observed the shrinkage of small lumen and expansion of the large lumen (see Fig. L5). We propose that this is the consequence of physical mechanism (Laplace law) at play. Exactly similar dynamics is reproduced in our model. In the experiment, hydrostatic pressure difference Δp across the cell layer (between lumen and ECM) balances with the tension of cell layer by the

Laplace law. This balance is attained immediately compared with the growth process of the lumen. Water influx through the cell membrane and cell-cell junction is mainly promoted by the unbalance between the osmotic pressure and hydrostatic pressure. This water influx drives lumen expansion.

In our computational model, ξ corresponds to the osmotic pressure of the lumen (see the analytical expression and simulation results Sec.2.6 in the Supplemental Information). It is proportional to the hydrostatic pressure. Growth of lumen occurs when the driving term ξ is larger than the opposing force from the cell layer which comes from exclusion volume interaction between cell and luminal fluid variable s in the Equation (8) in the Supplementary Information. If ξ is smaller than the force from cell layer, lumen will shrink and disappear. Due to the Laplace law, there exists critical size for the lumen growth. These effects are inherently included in the model. In a sense, our computational model is incorporating the lumen osmotic and hydrostatic pressure, tension from of the cell layer, expansion or shrinkage of lumen volume by the unbalance between osmotic pressure and the tension from the cell layer.

Additionally, talking about luminal pressure and mechanical cell interactions without actually checking the cell mechanical markers or mechanosensors weaken the study. At least the Rho/ROCK signalling, one of the main pathways mediating the cytoskeletal responses to mechanical signals or integrins should be analyzed. It would have been interesting to follow the spatial and temporal expression pattern of these markers to strengthen the study.

The actin/myosin localization and Rho/ROCK signaling pathways were studied in our associated articles Guyomar et al. BioRxiv and Mukenhirn et al, Developmental Cell, 2024. Briefly, they do not challenge the main conclusions of our results and we now report these features in the discussion.

2. The authors focused on only initial cell number to find the generic rules, but I believe that "cell number" is also one of the important factors. Why they did not count cell number during incubation time even Figure 2a visualized nuclei? Time-dependent cell number change in relation to cell cycle is much important to understand cell luminal pressure.

To clarify, we used initial cell number control to achieve reproducible cysts across systems with statistical significance. We employed an indirect method to capture these changes through the lumen index, a ratio of the lumen area to the overall size of the organoid, assuming cell volume remains constant.

This lumen index effectively accounts for cell number fluctuations. When lumen occupancy is high, fewer cells are present, while lower lumen occupancy implies a higher cell count. This approach indirectly reflects cell proliferation and its impact on lumen

dynamics, allowing for a consistent and reliable comparison across different systems without the need for direct cell counting at every stage.

Our measurements show that lumen volume and lumen occupancy increase with time. Higher initial cell number leads to larger lumen volume (Fig. L10).

Figure L10. Lumen index as a function of time. (a) volume of cysts as a function of time (b) as a function of initial cell number; volume of lumen as a function of time (c) and as a function of initial cell number (d). The lumen occupancy (e and f) plots show that lumen volume and lumen occupancy increase with time. Higher initial cell number leads to larger lumen volume and lumen occupancy. Linear fits in b, d, f.

3. There are lack of quantitative biological analyses such as a cell cycle, cell-cell

junction, cytoskelton, cell marker, and cell adhesion by gene and protein expressions. These biological analyses closely relate to cell proliferation, cell-cell meet, cell growth, and lumen formation.

Addionnaly, adding more statistical analyses will strengthen the results to support the raised correlations more robustly.

The goal of the study is to show the minimal physical rules mediating the nucleation and fusion of lumens across systems. In this context, we extracted the minimal physical parameters to group MDCK cysts and pancreatic spheres and distinguish with epiblasts. Future studies could incorporate more biological analyses, such as examining the cell cycle, cell-cell junctions, cytoskeleton, cell markers, and cell adhesion through gene and protein expression, would offer additional valuable insights.

4. Unfortunately, discussion of this manuscript is not enough. They just described only phenomena...I could understand "phenomena", but not understand the generic "rules". Also, consider discussing the implications of these findings for understanding organogenesis and morphogenesis across different developmental contexts to enrich the discussion section.

We appreciate your feedback regarding the need for a clearer distinction between phenomena and the underlying rules in our manuscript. In response, we have revised the text to better differentiate between the observed phenomena and the rules we derived from them. We also enriched the discussion section along the comments.

Minor comments:

1. English grammer should be improved by English native speaker.

The new version was reviewed by an English native.

2. Fig. 1: Please add cell umber changing against culture time.

We prefer to keep the same plot – initial cell number – since it is the main control parameter.

3. Fig. 1e: Statistica analysis and error bars are necessary.

We have added statistical analyses and more detailed explanations to the Figure legend. In Fig. 1e, each point represents an individual data point, while the dark line corresponds to the mean value, and the shaded region indicates the standard deviation. Additionally, we have zoomed in on the data and provided a separate, detailed plot in Supplementary Figure 2. This modification ensures improved clarity and statistical representation. The statistical tests are given when it is required.

4. Fig. 1c: Please discuss the reason why all samples even at different initial cell number reached to a "single" lumen.

We discuss further this point. Through the simulation studies of the present model, we found that the state with a single lumen surrounded by a cell monolayer is stably formed when the balance between cell proliferation and lumen growth is attained. This mechanism is explained in our paper (Tanida et al. Biorxiv, 2024).

5. Fig.1: Please show 3D reconstructed image of MDCK spheres. For the 16 initial cells it is not clear whether they really reached the single lumen stage.

We have now included a 3D reconstructed image of the MDCK spheres (Fig. L11.)

Figure L11 – 3D reconstruction of MDCK cysts, pancreatic sphere and epiblast. Scale bar: 50 μ m.

Also, analysis of the 16 initial cells reveals a decrease in the number of lumens over time. By day 8, most spheres had developed into a single lumen structure (see Figure L12a with multiple examples), with an average of 1.4 lumens per sphere. Further observations up to day 12 confirm that the spheres consistently exhibited a clear single lumen, as shown in the subsequent images (Figure L12b).

Figure L12 Phase contrast image of MDCK cysts that have reached a single lumen after 8 (a) and 12 (b) scale bar:50 μm .

6. Fig.2b,d: Please add cell number changing against culture time.

We prefer to keep the same plot – initial cell number – since it is the main control parameter.

7. Fig.2b,d: Statistica analysis and error bars are necessary.

We added statistical analysis and error bars.

8. Fig.2: Please show 3D reconstructed image of pancreatic spheres and epiblasts. Especially for the epiblasts whose lumen look quite flat and not really lumen.

We added these images (Figure L11).

9. Fig.2c: It is difficult to find lumen such as the sample of 1 of initial cell number at 48h. What is the threshold or criteria of lumen structure?

We define a lumen based on microscopy resolution by identifying the fusion of apical proteins into the junction. A "lumen" is characterized by a visible "hole" within the junction that is filled with apical proteins. To illustrate this criterion, we have provided a

supplementary Figure (see Fig. L13): a. MDCK doublet with a lumen in the middle. b. Top: Epiblast without a lumen. Bottom: Epiblast with a lumen. These supplementary images clarify the structural characteristics we use to define a lumen. They are included in the new version (Ext. Fig. 3).

Figure L13: Lumen Nucleation in MDCK cysts and Epiblasts. (a) Lumen formation is identified by the presence of a hole at the junction and the accumulation of the apical protein Podocalyxin. (b) Top two-panel row: Epiblasts showing accumulation of Podocalyxin without lumen formation, as no hole is present at the junction. Bottom two-

panel row: Lumen nucleation in epiblasts characterized by the accumulation of Podocalyxin and the presence of a hole at the junction. Scale bar: 10 μ m.

10. Fig.3a: There is no description evaluation method of cell cycle. Providing insights into how differences in cell cycle lengths might influence lumen formation dynamics would be beneficial.

We estimated the cell cycle based on published measurements and estimates from our own experiments. Both are consistent. The roles of cell cycle in lumen dynamics are analysed in our associated articles (Tanida et al, Lee et al.).

11. line 119: How to understand 10 cells in Fig.3f? I can recognize 7 nuclei in DAPI image. It is hard to understand cell number correctly from these images. Thus, 3D reconstructed confocal images are useful to understand cell number and location correctly.

We added 3D images to show how the cells count is obtained (Fig. L14). On average, we obtained 10 cells (Figure 2g).

Figure L14: 3D Visualization of Cell Counting in Epiblasts. The figure presents two examples, with the top row showing Example 1 and the bottom row showing Example 2. From left to right: 2D Cut: A cross-sectional view of the cells in 2D. 3D View with DAPI Staining: A 3D visualization highlighting nuclear structures with DAPI staining. Center of Cells in 3D: The central positions of cells in 3D space, marked in different colors to differentiate individual cells. Merged View: A composite image that integrates the 3D DAPI visualization with the 3D cell centers. Scale bar: 10 μ m.

12. Figure 4: Analysis of E-cadherin levels using only immunofluorescence measurement is not enough. PCR or Western Blot should be used to quantify and confirm this hypothesis.

We performed Western blots to compare E-cadherin levels between MDCK and epiblasts. We measured larger E-cadherin expression in MDCK cysts compared to epiblasts (see Fig. L2c.). This is inserted in the new version (Ext. Fig. 5c).

13. Figure 5: Clearly articulate the objectives of the simulations and how they complement experimental findings. The selection of the computational model should be more explained and discussed to understand the rationale behind choosing the phase field model.

Our aim was to test the rules of lumen dynamics observed in experiments. Therefore, we focused on tuning key physical parameters affecting lumen dynamics, such as lumen growth (ξ), cell division, and cell adhesion. To further validate this, we conducted new simulations to quantitatively study the effects of these parameters. In the revised manuscript, we added more readable explanation of the computational model for general readers as well as the rationale for choosing the phase field model in the Supplementary Information.

14. MDCK II cell line is from dog, while pancreatic spheres cells and embryonic cells both come from mouse, how these cells can also represent the human embryogenesis? This point or possible limitation should be discussed.

The strength of our approach is to show that rules can be conserved despite these variety of species. Our text is already long and for this reason, we did not include this point. Motivations for this are given in the new version of the Supplementary Note.

Reviewer #3 (Remarks to the Author):

The manuscript by Lu et al. describe similarities and differences when studying lumen formation by three different cell types. These are MDCK cells, E13.5 pancreatic epithelial cells and mouse embryonic stem cells (ESCs). Using these cell types, the authors wish to mimic lumen formation in kidney tubules, pancreatic ducts and blastocysts. They use fluorescently labelled cells, so that lumen formation can be followed dynamically over time. They also follow lumen formation by starting with only one cell per well and ending up with 16 cells per well. Lu et al. conclude that a small lumen first develops between adjacent cells (at least two cells in number) - they term this process "lumen nucleation". Subsequently, several of these small lumens coalesce to form a single large lumen in a process that they term "lumen fusion".

While these findings represent similarities between the three cell systems, they also found differences. Most notably, the cells around the lumen arrange differently: while ESCs form multiple layers around the lumen, the pancreatic cells and MDCK cells only form a single layer (but with the MDKC cells forming more what looks like a columnar epithelial cell layer while the pancreatic cells form a flat epithelial cell layer). Many of the images and image sequences are accompanied by quantification of multiple lumen parameters, and the authors also include simulations of lumen formation processes.

Finally, the authors manipulate their cell systems by injecting fluid into 1 of 2 lumens of MDCK cells or blocking cell connections by interfering with tight and adherens junctions. They observe that junctions are an integral part of lumen formation.

In general, the images are stunning, and the system to develop a lumen in a defined setting is straight. Still, the novelty of the presented findings is somewhat limited, given that the formation of a single lumen from small lumens have been described in many publications over the last decades and given that adherens junctions have similarly described as being essential for the lumen formation process. In the opinion of this referee, a substantial amount of work is needed to lift this manuscript with its beautiful images and quantifications/simulation to a stage of innovation that is needed for publication in a top journal.

The point of our study is not to show small lumens effect and adhesion roles but rather to group phenomena in single effective framework from the physics of lumen. If we understand the comments from the reviewer from the biology of lumen, we defend that the physics of lumens needed a new study in its own right and this is the main novelty of our approach merging theoretical physics, quantitative experimental physics and cell biology.

The scope of our study is to show that comparison between systems can be explained with the same physical framework. In short, we present nucleation and fusion mechanisms, quantified and tested with physical models, and validated through perturbation experiments. In the revision, we further show the difference of hydrostatic pressure (Fig. L15.).

Overall, our study represents an innovative approach to comparing lumen kinetics, highlighting the connection between cellular and tissue mechanics with lumen dynamics.

We included modifications mentioned below in the new version of the MS.

Figure L15: Estimation of Hydrostatic Pressure in MDCK Cysts and Epiblasts. (a) Lumen volume changes in MDCK cysts and epiblasts measured through lumen drainage induced by laser cutting. Images show midplane cross-sections of cysts before and after laser cutting, with red lines indicating the positions of the cuts. (b) Estimation of hydrostatic pressure in the lumen using the Hagen-Poiseuille law (refer to Mukenhirn et al., 2024, and the Methods section for further details). (c) Lumen volume changes over time following laser cutting. The dark line represents the mean volume change, while the shaded region indicates the standard deviation. (d) Hydrostatic pressure measurements show an average of approximately 65 Pa in MDCK cysts and close to 0 Pa in epiblasts. Data are based on three independent experiments ($N=3$), with 27 MDCK cysts ($n=27$) and 22 epiblast samples ($n=22$). Scale bar: 10 μm .

Major points:

(1) The authors work on cysts rather than tubes, the latter being the prevailing lumenized structures in multicellular organisms. In this way, only the ESC work has a strong physiologic counterpart *in vivo*. Thus, the authors must try to get cell cords as a starting point to develop physiologic kidney tubules and physiologic pancreatic ducts rather than develop kidney cysts and pancreatic cysts *in vitro*. Given that they work entirely on an *in vitro* system with no evidence that their findings are relevant for the situation *in vivo*, they must either develop tubes *in vitro* and/or study lumen formation *in vivo* to validate their *in vitro* findings on cysts.

We respectfully disagree with the reviewer's point that our study lacks relevance due to its focus on cysts rather than tubes. Early organogenesis indeed begins with the formation of micro-lumens, followed by lumen fusion and eventual patterning into tubular structures such as those found in the kidney and pancreas (Villasenor A et al, 2010. Kesavan G et al. 2009). Thus, the processes of lumen nucleation and fusion that we study are intrinsic components of organogenesis.

However, we acknowledge the reviewer's suggestion to test our findings within tube-like structures. In response, we redesigned our wells to simulate a tube-like environment.

As a result, we observed that cysts could follow the shape of the tube while still undergoing lumen fusion into a single lumen (Fig. L16). Additionally, the dynamics of lumen nucleation were conserved, with the number of lumens increasing proportionally to the initial cell number.

These findings confirm that the fundamental rules of lumen nucleation and fusion are applicable regardless of the well design. Therefore, our *in vitro* cyst models provide valuable insights into the processes underlying organogenesis and can be relevant to understanding *in vivo* phenomena.

Figure L16. Extrinsic control of MDCK cyst growth using micro-Well confinement. (a) Comparison of lumen nucleation in circular and rectangular micro-well shapes with varying initial cell numbers. Podocalyxin is labeled in red, and E-cadherin is labeled in green. (b) Microengineering-based approach for controlling the shape of MDCK cysts. Representative images show cyst morphology at Day 0 (left) and Day 4 (right). scale bar:20µm

(2) The role of pressure inside the lumen is speculative. The whole lumen fusion process could be explained by reducing surface tension. As a matter of fact, many tubes (such as blood vessels) are leaky when forming a lumen, meaning that an increased pressure inside the lumen cannot be a generic rule for lumen formation. Here the authors must completely revise their model or at least experimentally work on an alternative model and discuss it.

To answer this point, we compared the surface tension between cysts with multiple lumens and those with a single lumen, finding no significant differences in surface tension between the two configurations (see Fig. L17).

Regarding the role of pressure in driving lumen fusion, we agree with the reviewer's speculation. Our findings support the idea that fusion mechanisms in both MDCK cysts and pancreatic spheres are driven by lumen expansion rather than increased internal pressure (see Fig. L4 and 5). Quantitative analysis and imaging indicate that smaller lumens tend to empty into larger lumens during the fusion process. Additionally, we observed that laser ablation resulted in faster fluid flow from smaller lumens compared to larger ones.

In contrast, during the fusion process in epiblast cells, there was no significant change in lumen size. The epiblast system is leakier (Kim, Y. S. *et al.* 2021), and we noted that lumen fusion is correlated with directional cell movement. Furthermore, inhibiting cell motion with blebbistatin effectively halted the fusion process.

These observations underscore the complexity of lumen formation mechanisms and suggest that while pressure may play a role in certain contexts, the dynamics of lumen fusion are influenced by various factors, including surface tension and cellular motion. We expand our discussion to report these important points in our revised manuscript.

Extended Figure 7

Figure L17. Surface tension measurement of MDCK cysts with single and multiple lumens. (a) Brightfield Images: Representative brightfield images show MDCK cysts undergoing surface tension measurements, highlighting cysts with multiple lumens (top panel) and a single lumen (bottom panel). Scale bar: 20 μm . (b) Box plot analysis: Data are presented as mean \pm standard deviation (s.d.) with individual data points overlaid, providing a view of both central tendency and data variability.

Minor points:

(1) The authors do not walk the reader through their Figures, but often skip mentioning Figure panels in the main text, which must be changed.

We modified.

(2) The interpretation that pressure differences may explain the differences between the lumen indices is likely wrong and does not pay any attention to the simple fact that the morphology of the different epithelial cell layers (flat, columnar and multi-layered) can (at least in part) explain these differences.

In our study, lumen indices were defined as the size of the lumen relative to the size of the organoids. This metric inherently reflects cell shape, as higher lumen indices correspond to thin monolayers composed of flat cells. The shape of the cells, in turn, results from the balance of mechanical forces acting on them (see our preprint [25] for instance).

Additionally, we measured luminal pressure across organoids with varying lumen indices and found a positive correlation (Fig L1). Thus, while luminal pressure is an important factor, we agree that cell shape also serves as a critical indicator of lumen index differences.

(3) In Figure 6a, dextran is not visible. No quantification is provided. The authors are advised to transfer the lumen fluid from one MDCK cyst to another, as to avoid that the fluid composition injected (with potential differences in ion concentrations) triggers changes in lumen development.

To clarify our methodology, MDCK cysts were initially grown in medium without dextran. We then injected medium containing dextran, which is visible in the third column of Figure 7a—showing images before (top) and after (bottom) the injection. We performed quantification of lumen volume changes resulting from this injection.

As per the reviewer's advice, we injected dextran into one lumen, and the subsequent increase in lumen size facilitated lumen fusion (see Fig. L18). We checked that these details are clearly presented in our revised manuscript.

Figure L18: Micropipette inflation swelling experiments on MDCK cysts with two lumens. The left two panels show the cysts before injection, and the right two panels show the cysts after injection. Each row represents an individual example (four examples total). Cysts are labeled with E-cadherin (green), Podocalyxin (red), and Dextran (magenta). The inflation procedure triggers lumen fusion. Scale bar: 10 μ m.

(4) Given that the authors reveal differences between the cysts they develop in vitro from three different cell types, can they convert MDCK cysts into multi-layered epiblast cysts and vice versa by experimentally manipulating these cysts? At least, this would be more innovative than just showing that junctions are needed for proper lumen formation.

In our revised analysis, we included a comparison of cell shapes between MDCK ZOKO cysts and epiblast cysts. We found that both exhibit similar morphologies, characterized by small lumen occupancy and elongated cell shapes (Fig. L19). We also show similar fusion events (Fig. L19).

Figure L19. Comparison of morphology between epiblast and MDCK ZO-KO cysts. Left: Epiblast cyst. Right: MDCK ZO-KO cyst. Both cyst types exhibit similar cellular and tissue morphology, as shown by the staining of Podocalyxin (red) and E-cadherin (green). Scale bar: 10 μ m.

(5) The authors speculate about an ion pumping mechanism involved in coalescence of small lumens. Which ion pump do they think of? Again, this referee is not convinced about pressure differences driving major changes in the developing lumen. If they insist though, they must use KO cells for a specific ion pump and provide evidence for its role in lumen formation in their system.

In our study, we focus on the speed of lumen expansion as a critical factor. We do not claim that ion pumping alone can fully explain lumen size changes. For instance, epiblast cells function as a leaky system, which cannot sustain rapid increases in lumen size (Fig. L2d and Fig. L20). We added pressure perturbation experiments by fluid injection as a new experiment addressing the role of pressure.

The physics of lumen involves a balance between hydrostatic pressure and tissue elasticity. We show in this paper and in the associated articles that this vision is relevant.

We thank the reviewer's insights and we modified the text to clarify these concepts.

Reviewer #4 (Remarks to the Author):

This manuscript from Lu and colleagues proposes generic rules of lumen nucleation and fusion in epithelial organoids. The number of lumens, and lumen fusion, are observed in MDCK cells, pancreatic cells, and mouse derived embryonic stem cells as a function of cell number and culture time, with the observation of multiple nucleation events and an initial increase in lumen number in larger cell clusters followed by cell fusion. In MDCK and Pancreatic clusters, lumens form during cell division and cell

meeting events, whereas lumen nucleation occurs in rosettes for the mESCs. Lumen fusion in MDCKs and pancreatic spheres is associated with increase in lumen index, whereas lumen formation in the mESCs is associated with cell rearrangements. Simulations are performed of nucleation and fusion based events for the three systems. Inflation via a pipette promotes fusion events in MDCKs, and decreased cell-cell adhesion (via molecular agents or knockout) promotes fewer lumens in MDCKs. The authors conclude that fusion dominated by increase in pressure in pancreatic and MDCK spheres whereas epiblast lumens fuse by cell motion.

The study of lumen fusion is quite interesting, and there are some nice elements to this study, in particular the control of starting cell number and the detailed monitoring of fusion. However, there are a number of major concerns that dampen enthusiasm for the manuscript and its suitability for Nature Communications in its current form. Mainly, there are limited mechanistic insights resulting from this largely observational study, and some of the conclusions are currently not supported.

We thank the reviewer and we address the associated pending points below. Our study is not just an observational study but we did multiple perturbations to test mechanisms.

Comments:

1. Physiological relevance: One point of confusion is the relevance of these lumen fusion events for biology. One would expect likely the 1 cell case to be most relevant in this regard. This should be explained so the reader can better understand the physiological relevance of the observations.

We understand the reviewer's concern about the physiological relevance of lumen fusion events, particularly in relation to different starting cell numbers. We would like to clarify that our study's focus on lumen dynamics is rooted in the principles of organogenesis. It is worth noting that '1 cell case' is not necessarily 'the most relevant' in development where cell is rarely alone.

In mammalian development for pancreas, organogenesis typically begins with clusters of cells that initially form micro-lumens. These micro-lumens then undergo fusion to develop into continuous tubular structures. Thus, the processes we observe—lumen nucleation, fusion, and expansion—are fundamental steps in this developmental sequence (Villasenor A et al, 2010. Kesavan G et al. 2009).

The observation of lumen dynamics starting from a single cell, as well as from clusters of cells, provides valuable insights into the early stages of organ formation. Our methodology offers a novel approach by quantitatively controlling the initial cell number and systematically comparing lumen dynamics. This innovative method allows us to isolate and analyze the fundamental mechanisms driving lumen formation, offering a clearer understanding of the process from a developmental biology perspective.

2. Mechanistic insights: The studies are largely observational, observing lumen

nucleation and fusion, with only the inflation studies and e-cadherin perturbation studies pointing towards some mechanistic insights. I do not believe the authors have supported the conclusion that “lumens fuse by an increase in lumen volume for pancreatic spheres and MDCK cysts, whereas cell convergent directional motion leads to lumens fusion in epiblasts.”

We added more experimental evidence and together with former results, we further substantiate and confirm our claims. They are inserted in the new version of the MS.

First, we quantitatively compared the three systems in terms of lumen size, cells motion. We observed that small lumens empty into large lumens for both MDCK cyst and pancreatic sphere. This indicated that hydrostatic *pressure* could explain fusion in MDCK cyst and pancreatic sphere. To further support this point, now we also add pressure measurement with laser cutting. We find that, small lumen has a higher pressure than the large lumen in cysts with 2 lumens. In addition, we confirmed the roles of adhesion with perturbation experiments, injection experiments and cadherin KO experiment. All these experiments further supported our claims are shown to be relevant.

a. Just because artificial inflation promotes fusion doesn't necessarily mean that pressure is driving fusion in the natural structures. For example, I would anticipate that inflation would drive lumen fusion in the mESC as wells. That study should be performed.

We performed this experiment and it is reported above (Fig. L20). Epiblasts were dissociated with injection experiments and it suggests epiblast could not sustain this volume change. Together with our pressure measurements, this supports our description of epiblasts. This is inserted in the new version of the MS.

Figure L20. Micropipette inflation swelling experiments on epiblasts with two lumens. Top panels show the condition before injection, and bottom panels show after injection, with three images corresponding to (left to right): dextran staining, SiR-actin + dextran overlay, and transmission light microscopy

Further, individual lumen size should be tracked prior to fusion (this may be related to lumen index but this is not clear).

Indeed, we have tracked individual lumen sizes to illustrate lumen dynamics prior to fusion. This aligns with your points. To clarify, we have plotted the lumen index, which is defined as the ratio of lumen area to cyst area. We have also included a more detailed explanation of this measurement in the methods section. In addition to Figures 4 b and c, we have introduced Supplementary Figure 4, which offers a more comprehensive view of lumen behavior across multiple replicates for more robustness.

Finally, natural lumen expansion should be inhibited (e.g. via increased osmotic pressure) to further support this conclusion.

To further investigate how natural lumen expansion may be influenced by external factors such as osmotic pressure, we conducted experiments using osmotic shock to assess its impact on lumen formation. Specifically, we applied 300 mM D-Mannitol to MDCK cysts starting with 8 initial cells on day 1.

Our results (see Figure 7b and c) indicate that osmotic shock significantly affected lumen formation. Under osmotic stress, cysts predominantly formed a single lumen, while control cysts, without osmotic treatment, typically developed multiple lumens by day 3, averaging around 3 lumens per cyst (Figure 7b and c).

These findings suggest that osmotic changes can modulate lumen formation, highlighting the critical role that mechanical and osmotic pressures play in regulating lumen dynamics.

b. That cell-cell junctions are a barrier to lumen fusion would not be surprising. However, the data in Fig. 6e appear to show that these various perturbation reduce initial lumen nucleation and not necessarily lumen fusion. This needs to be analyzed and clarified.

To address this point, we noted that fusion events also occurred during the nucleation phase. Therefore, we compared the rate of fusion (fusion slope) between conditions. We observed a steeper fusion slope in cadherin KO cyst, indicating a faster decrease in the number of lumens compared to the WT groups (Fig. L21). This suggests that lumen fusion is also modulated by adhesion.

Figure L21: Comparison of fusion slopes between MDCK WT and Cadherin-KO Cysts. The fusion rate of lumens was analyzed using cysts starting with 16 cells as the initial number. Changes in lumen count were tracked from day 1 to day 2, comparing MDCK WT with Cadherin-KO cysts.

c. The conclusion on convergent directional motion leading to lumen fusion in “epiblasts” is based on correlative/observational data, and its not clear what the driving factor is. For example, it could be that cell rearrangements are passive and the consequence of pressure-driven fusion events. To show a causal relationship, the cell movement would need to be modulated and the impact on fusion assessed.

To test this point, we incubated epiblasts with blebbistatin: motion was inhibited and fusion was arrested (Fig. L22).

Figure L22. Perturbation of lumen fusion in the Epiblast. (a) Representative snapshots of the lumen fusion process in the epiblast. The top panel shows the control condition treated with DMSO, while the bottom panel displays the effect of Blebbistatin treatment. Lumen are visualized using SiR-actin (gray). Scale bar: 5 μm .

d. At a fundamental level, its unclear what is driving the hypothesized pressurization and convergent directional motion. The authors could examine the role of osmotic pressure for example (image ions, perturb ion channels, and increase osmotic pressure in media), and the role of contractility (imaging of the contractile actomyosin network, and inhibition of its activity). Substantial mechanistic studies are needed to clarify the molecular drivers of fusion in these different contexts. Further, cell division could be perturbed (i.e. with an agent like mitomycin C) and the impact on cell division and fusion events monitored.

To support our conclusion, we compared lumen size between three systems. We used micropipette inflation approach to increase lumen size. The increase in lumen size leads to fusion of lumens in MDCK cysts. However, in a similar injection experiment on the epiblast, the entire epiblast simply collapses with the significant increase in lumen size by injection. In line with your suggestion, we further performed osmotic shock on MDCK cysts using 300 mM D-mannitol. Altogether, we captured the physical parameter at play in setting the luminogenesis and we report a consistent picture.

Results modulating cell division are reported in another paper of our network our network (Lee et al, BioRxiv, 2024) and they are consistent.

e. Related to the pressurization question, it was not obvious to this reviewer what the significance of the LI parameter was. I am wondering whether solidity would be a better measure of the role of pressure, as described in a manuscript by Vasquez, et al., Nature communications 2021 (“Physical basis for the determination of lumen shape in a simple epithelium”). Further, lumen area alone could be shown.

The Lumen Index (LI) is defined as the ratio of the lumen volume to the overall volume of the organoid (Tanida S et al, 2024). A larger LI indicates that the cells within the organoid are more stretched, suggesting higher internal pressure, while a smaller LI implies that the cells are more elongated, reflecting lower internal pressure. LI effectively captures the effects of osmotic and hydrostatic pressures, which are crucial factors in determining lumen dynamics. This index provides a direct measure of how pressure influences the shape and volume of the lumen.

In contrast, solidity is a measure that quantifies the proportion of the lumen area relative to its convex hull area. While solidity can provide insights into the overall shape and potential pressure effects on the lumen, it primarily relates to local forces such as tension. As noted in the manuscript by Vasquez et al. (Nature Communications 2021), solidity is valuable for understanding the physical basis of lumen shape but might not fully capture the broader implications of pressure dynamics.

Furthermore, as the solidity index characterizes convexity/concavity of the lumen shape, our phase field model (PFM) can reproduce both convex and concave apical surfaces which is generally not possible for the standard vertex model for instance. Thus, our model has advantages to capture the convexity or concavity, but we employed LI rather than the solidity. From our simulation results, convex apical area (high solidity) just corresponds to lower pressure in the lumen and concave apical area (low solidity) corresponds to higher pressure in the Young-Laplace law.

3. It was difficult to follow the model and what it adds to the paper. If lumen nucleation and volume, and each of the observable parameters is part of the input assumptions, it is not surprising that the authors capture behavior of the different systems, especially with a different model used for the epiblasts. What is the specific question that the model is trying to address, and how is it validated? What are the input assumptions, how are the parameters estimated, and what are the constraints? This section should be expanded/clarified and these questions addressed.

We revised the model description in the Supplementary Information (SI). The specific questions addressed from a theoretical perspective are described in Section 1 of SI. Input assumptions, validation of the model, and choice of parameters as well as the advantages of each model are described.

In short, our model is a mechanical model. Each cell and lumen evolve in time according to the gradient of associated energy. Interactions among cells, lumen, and ECM are included in the interaction energy. We set initial conditions and parameters,

after that time evolution is spontaneous. This implies that the time evolution of cell growth and division, lumen nucleation, fusion, and growth occur by self-contained dynamics and force balance. Therefore, it is not obvious if the resulting dynamics reproduce the experimental results. If so, that strongly supports that the simple mechanical model captures underlying mechanism of the development of the cysts.

In addition, lumen driving force λ_i in our model is comparable to the osmotic pressure. As shown in the SI of the revised manuscript, phase field model the relation between osmotic pressure, tension of cell layers, and pressure difference (Laplace law) holds. Therefore, with minimum input assumptions, the model performs self-contained dynamics of the growing process of the cysts. All parameters in the model are non-dimensionalized. Thus the model allows mostly qualitative comparisons between experiment and simulation, but since factors such as lumen growth driving, cell-cell adhesion, and cell proliferation are incorporated in the model, exploring the relative strength of these parameters enables us for theoretical predictions.

Minor comments:

-The term epiblast is likely inappropriate. At best, this is a model of the epiblast, though its not clear how good of a model. This should be clarified and the use of some qualifiers should be included.

We adopted the term "epiblast" based on its established use in the field, as referenced in studies by Shahbazi et al. (2017) and Bedzhov & Zernicka-Goetz (2014).

-Fig 3a – difficult to understand what are the data and what is the fit. Also, the R2 value, or other appropriate statistic for fit should be included).

In Figure 3a, we normalized the nucleation rate by the cell doubling time. This normalization adjusts the time scale to the number of cell cycles, which directly correlates with the number of cells. This provides a straightforward method to account for differences in cell proliferation rates across conditions.

To clarify the data presentation, we have now included both the raw and normalized data, offering a clear comparison of the results before and after rescaling (Figure L3). Additionally, we have added the R² value and details on the fitting method in both the figure legend and the methods section, see also the new version of the MS.

-Fig. 3g – what about the multi-lumen clusters?

In this analysis, we focused on single-cell initial conditions to characterize lumen nucleation, where the lumen typically forms at the center of the rosette structure. However, as seen in multi-lumen clusters (Figure 2c), when starting with a higher initial cell number, such as 16 cells, multiple rosettes form, each with a lumen at its center. This demonstrates that with larger clusters, multiple lumens can nucleate simultaneously within distinct rosette centers.

-Fig. 4c – is this averaged data? Are there statistics here.

We have plotted individual examples to illustrate lumen dynamics, which also aligns with your subsequent questions. However, we acknowledge that additional biological replicates are necessary. To address this, we have included Supplementary Figure 4, which incorporates further replicates. In addition, we compared 3 systems together, in which the dark line corresponds to the mean value, and the shaded region indicates the standard deviation (see above Figure L4).

-6b/c – do lumens also leak out into the surrounding extracellular space in this experiment?

The experiment consists of two distinct phases:

Initial phase: During this phase, the lumen volume remains stable, with no significant changes observed. There is no clear evidence of leakage of luminal contents into the surrounding extracellular space.

Later phase: Over time, a reduction in lumen volume is observed, along with instances of lumen fusion. These changes suggest the possibility that luminal contents may either be reabsorbed by the cells or released into the extracellular space. Additional analysis is required to determine whether the observed reduction is a result of extracellular leakage, cellular uptake, or luminal collapse.

-Inflation studies require some kind of quantification ideally and statistical tests to make stronger conclusions.

We performed quantification of lumen volume changes resulting from this injection and statistical tests.

Following a reviewer advice, we injected dextran into one lumen, and the subsequent increase in lumen size facilitated lumen fusion (see Fig. L18). These elements are presented in our revised manuscript.

Reviewer #5 (Remarks to the Author):

In this article, Linjie Lu and colleagues aim to identify the common dynamics in lumenogenesis across three different systems: MDCK cysts, pancreatic spheres, and epiblast cysts. This unifying observation is achieved through the integration of quantitative cell biology, microfabrication techniques, and theoretical frameworks. In this sense, the authors firstly report that the maximum number of lumens is dictated by the initial cell count, without influencing the steady state, resulting in a final single lumen. They then describe two phases: a nucleation phase followed by a fusion phase. Lumens

emerge between two cells in pancreatic and MDCK cysts during the nucleation phase, while in epiblasts, they form at the rosette stage between ten cells. During the subsequent phase, lumens in pancreatic spheres and MDCK cysts fuse through an increase in lumen volume. In contrast, in epiblasts, the fusion results from the convergent directional motion of cells. To support these findings, the researchers replicated the dynamics of lumens numerically using a phase field model that incorporates basic rules for cell proliferation, cell adhesion, and lumen growth. By manipulating cell adhesion and lumen volume in MDCK cysts, the authors were able to successfully replicate the fusion dynamics observed in pancreatic spheres and epiblasts.

This study employs a comprehensive approach by investigating lumen dynamics across multiple cellular systems (MDCK cells, pancreatic spheres, and mESC-derived epiblast organoids). This breadth allows for a comparative analysis, enhancing our understanding of common principles underlying organogenesis. By combining experimental techniques such as cell tracking with numerical simulations, the study provides a robust framework for studying lumen formation. This integration allows for the validation of experimental findings and the exploration of underlying mechanisms in a controlled computational environment. Indeed, we are presented with a groundbreaking mathematical simulation that appears to integrate the three models discussed in the article. Also, The study includes experimental perturbations to validate the observed mechanisms of lumen formation. By manipulating factors such as lumen growth rate and cell-cell adhesion, the researchers provide empirical evidence supporting their computational and theoretical predictions.

Overall, the study's strengths lie in its comprehensive approach, integration of experimental and computational methods, quantitative analysis, identification of both differences and commonalities across cellular systems, experimental validation of findings, and potential implications for tissue engineering. These strengths contribute to a deeper understanding of lumen dynamics in organogenesis and pave the way for future research in the field.

We thank the reviewer for the positive evaluation.

However, While the article presents a comprehensive study on the dynamics of lumen formation in organoids, there is limited new information or groundbreaking discoveries presented in the research. The study largely confirms existing knowledge and observations regarding organogenesis and lumen formation. It acknowledges previous research on the subject and primarily seeks to validate and compare these findings across different cellular systems. Indeed, Many of the mechanisms and processes described in the study, such as the role of cell proliferation, cell-cell adhesion, and luminal pressure in lumen formation, are well-documented in existing literature. The study does not introduce significantly novel mechanisms or pathways involved in organogenesis. Hence, given that it is widely recognized that various facets of lumen formation are common across these models, the presence of a biphasic behavior is not surprising (2). This commonality also extends to the acknowledged roles of cellular junction remodeling and luminal ion pumping.

New pathways or molecular mechanisms are not the target for this study. We reformulate the article to highlight that the physics of lumen is the point under tests, and we explain in details its novelty.

In addition, aside from the mathematical simulation, the authors introduce a compelling point by highlighting that in pancreatic spheres, the lumen index surpasses that of MDCK cysts, and fusion is likely driven by luminal ion pumping. In contrast, MDCK relies on cell-adhesion forces, while epiblast fusions are primarily governed by cellular motion. To improve clarity, it would be beneficial for the authors to explicitly explain each dominating mechanism for each model throughout the entire text, instead of only in the discussion section.

We agree with the reviewer and we explain in more details in the new version.

In sum, while the study may provide some additional insights into the specific dynamics of lumen formation in different cellular systems, the findings are largely incremental rather than transformative. The observed differences in timing and mechanisms of lumen formation across systems, while interesting, may not represent a significant departure from existing knowledge.

Our study aims at integrating different systems within the same description of lumens. The scope is now presented more directly in the revision.

Minor points

1.) Fig. 4B: Sir-Actin is used as staining for epiblast cysts but phase contrast for pancreatic spheres. To improve image quality and clarity, I would suggest the use of Sir-Actin in pancreatic spheres too.

To enhance image quality and clarity, we conducted additional experiments using Sir-Actin staining on pancreatic spheres as well (see above Fig L4a). The results demonstrate a comparison between phase contrast images and Sir-Actin staining. Our findings indicate that faster lumen fusion, as visualized by Sir-Actin staining, leads to the fusion of the lumen in pancreatic spheres. This approach provides a more detailed visualization of the cellular structures and processes involved.

2.) General review of English language is advised.

The text of the new version was read by an English native.

References:

1.) Carleton AE, Duncan MC, Taniguchi K. Human epiblast lumenogenesis: From a cell aggregate to a luminal cyst. Semin Cell Dev Biol. 2022 Nov;131:117-123. doi:

10.1016/j.semcdb.2022.05.009. Epub 2022 May 27. PMID: 35637065; PMCID:
PMC9529837.

We added this reference.

REVIEWER COMMENTS

Reviewer #1 (Remarks to the Author):

The authors have managed to address our queries in the rebuttal and they have done a lot more experiments to support their claims now. They also changed to a more appropriate title and in terms of the work done so far, it is quite complete for their claims.

We thank the reviewer for his/her constructive comments and for his/her recognition of our extensive work to address successfully the queries, with many additional experiments and completeness.

The irony is that the title also accurately reflected how narrow the work is. Probably only the motion-directed lumen fusion in epiblast is relatively novel.

This reviewer acknowledges in this second review our convincing ways to support our claims and asked for “more work” so that the MS can be “considered for publication” in the first review. We have integrated lumenogenesis across 3 systems in a coherent and comparative way to our knowledge for the first time. Taken together, this suggests that novelty is now substantiated for publication.

The pressure measurements appear sound, although a value of close to 0 Pa for the case of epiblasts may suggest that the approach to measure pressure may not be sensitive to measure very low pressure.

Our measure is within the error and the limit of the method. Therefore, we give ~0 Pa as an estimate bearing in mind that more resolved methods would be needed to obtain the specific value for the hydrostatic pressure of the epiblasts. This does not affect our conclusions.

Specifically, we obtained 65 Pa for the MDCK cyst. As explained in modified Figure 6, this hydrostatic pressure value is known to be high enough to open the lumen with a known cortical tension value. However, when it is much smaller than 30 Pa, including 0 Pa, the lumen cannot grow until the number of cells in the rosette exceeds about 10 cells. In that sense, our pressure measurement is sufficient to distinguish these differences as explained now p. 5.

The tension measurements are not adequately described in the main texts nor the methods.

We now insert more detailed parts for the tension measurements p. 11.

Also, in the revised manuscript there are still mistakes like missing mention of sub figures in figure legends and referencing the wrong figures in the main text.

We scanned now the text again and we mention systematically all figures and subfigures.

Reviewer #2 (Remarks to the Author):

The authors addressed nearly all technical and conceptual concerns raised in the initial review, providing additional experiments and data analysis to validate their claims and a more detailed discussion of the results' implications.

We thank the reviewer for the positive evaluation and for appreciation of the extensive work to support the claims.

However, there is still a lack of mechanistic depth and an overreliance on observational data. Here are the aspects that still need improvements:

- About the simulation of luminal pressure: The authors' explanation is reasonable. However, the connection between ξ and pressure could be more explicitly tied to the observed data.

We add now the detailed explanations for the connection between ξ and pressure in the Supplementary Information section 2.6.. Shortly, we provide quantitative comparison in two companion papers, Tanida et al [26] (revised) and Lee et al [23]. We obtained 65 [Pa] for MDCK cysts. We can compare this value with the simulation parameters. In Fig. 6b, simulation of MDCK cysts was performed at $\xi = 0.18$ which corresponds to 180 [Pa] for osmotic pressure (see the Table 2 in the revised Tanida et al. [26]. Cavity pressure in the simulation corresponds to $\Delta p = 0.7 \times 180 = 126$ [Pa] which is twice the observed pressure. This twofold difference in this model is satisfactory.

It would be better to add clarification on whether the model's assumptions (e.g., constant osmotic pressure) align with biological variability.

We follow this assumption for constant osmotic pressure already used by Chan et al. *Nature* 571, no. 7763, 112-116 (2019). Shortly, osmotic pressure, and cavity pressure reach almost constant values before cell proliferate. We add this point in the Supplementary Theory.

The assumption that the osmotic pressure is constant for the growing lumen can be justified if the lumen volume and osmotic pressure reach a steady state quickly enough in each state defined with its cell number. Let's estimate the timescale required for the lumen volume and osmotic pressure to reach a steady state based on the literature. A steady state of the lumen volume and osmotic pressure (which is determined by the ion concentration in the lumen and the cell) is achieved when active pumping and leakage of ions are balanced, and the time required to reach a steady state is determined by leakage. The leak constant is reported as $\lambda = 3 \times 10^{-3} [\mu m. s^{-1}]$ with biological variability (Torres-Sanchez, 2021).

This gives a time scale, $\tau \sim \left(\frac{R}{3\lambda}\right) = 3 - 3000$ [sec], which is much faster than the cell doubling time (~24 hours). This means that even if the cyst is growing dynamically, osmotic pressure and cell volume will reach a steady state quickly enough after each cell division, which supports calculation for each cell number.

Detailed calculation is shown in Supplementary Information 2.8.

Also, please show how hydrostatic pressure differences actually correlate with lumen size and shape, to validate the model as following the Laplace law.

We have included already this correlation in the Suppl. Fig. 6a and b.

Same can be done for hyperosmotic or hypoosmotic solutions (e.g., adding D-mannitol or sucrose to the medium to perturb osmotic gradients and monitor changes in lumen size, fusion rates, and dynamics). Alternatively, the authors can also use inhibitors of ion channels or pumps to disrupt osmotic pressure regulation and again observe the effects on lumen growth and fusion.

This point was already included in the Figure 7 and we report that the speed of lumen fusion is increased by the change of osmotic pressure by addition of D-mannitol. We also had explained the limitation of MDCKII cell line because of the absence of CFTR receptor which prevents the use of the classical forskolin for example.

- Impact of cell number on luminal pressure and dynamics: The cell number was measured indirectly using the lumen index and data correlate only initial cell numbers with lumen dynamics. While the authors' explanation seems logical, it still lacks direct evidence correlating cell number changes during culture with pressure. A more detailed time-course data on cell proliferation or more justification for their reliance on indirect measures is needed for this study. At least the final real cell number should be provided.

To address the comment of the reviewer, we measured the cell number over time together with lumen occupancy – see below panel B. The linearity for the three systems shows that our correlation remains valid for cell number as well. This comes together with the assumption that proliferation is regular.

Figure L1a illustrates the lumen volume as a function of total cell number. Pancreatic spheres exhibit a larger lumen volume compared to MDCK cysts and epiblast models. **Figure L1b** shows lumen occupancy as a function of cell number, confirming the fact that pancreatic spheres and MDCK spheres have a linear dependency with cell number.

The correlation matters primarily with initial cell number. The successful comparisons for this readout for experiments and for simulations show that initial cell number is already sufficient to identify mechanisms for lumenogenesis. The additional support provided now for the total cell number adds another confirmation to our mechanisms.

- *Lack of biological analysis: It is understandable that future studies will integrate these biological factors. However, the current study relies mostly on images analysis and more quantitative data are needed to enhance the robustness of the conclusions by mechanistic studies. The lack of molecular and biological analyses currently limits the study's broader biological relevance and the relevance for a publication in this journal. The authors should add immunostaining for mechanosensitive markers (e.g., YAP/TAZ nuclear localization, Rho/ROCK signaling, actomyosin organization) as well as RNA or protein expression analysis of aquaporins or ion transporters involved in maintaining osmotic gradients. Alternatively, inhibiting actomyosin contractility with drugs like blebbistatin or Y-27632 (ROCK inhibitor) can be done too.*

To address the comment, we performed new experiments – see Figures L2 and L3 below : (i) immuno-fluorescence for actin, myosin and E-cadherin and we report that myosin is enhanced at the apical side in epiblast – see also our companion paper for actin and myosin localization (Guyomar et al., BioArxiv <https://www.biorxiv.org/content/10.1101/2024.08.22.609097v1>); (ii) myosin localization during fusion does not change for MDCK spheres; (iii) we tested inhibitors at typical concentrations and incubation conditions, blebbistatin which did not affect fusion, and latrunculin A which did not prevent fusion but led to changes in lumen volume. We had addressed this interplay between active forces and lumenogenesis in other articles

article (Lee et al., BioArxiv

<https://www.biorxiv.org/content/10.1101/2024.05.29.596462v2>; Guyomar et al. BioArxiv

<https://www.biorxiv.org/content/10.1101/2024.08.22.609097v1>). However, we insist on

the fact that further elements would go beyond the scope of this article. Altogether we

report the main results in the new version p. 6 and in Fig. 8 and in Ext. Fig. 12 and in

Suppl. Note p. 12.

Figure L2 (a) Comparison between MDCK cyst and epiblast models using immunostaining. The readouts include F-actin (green), myosin (red), and E-cadherin (magenta). These typical images show the accumulation of myosin at the apical side for epiblasts. (b) Dynamics of myosin during the lumen fusion process, showing no major changes in either myosin intensity or localization during fusion events.

Figure L3: Inhibition of myosin during MDCK cyst fusion. *a*) Representative images of MDCK cysts cultured under different conditions. The top panel shows cysts treated with DMSO (control), while the bottom panel shows cysts treated with 10 μ M blebbistatin to inhibit myosin activity. Scale bar: 5 μ m. *b*) Quantification of the number of lumens in MDCK cysts treated with DMSO and blebbistatin. The analysis reveals no significant differences in the number of lumens between both conditions. Blebbistatin was added at Day 2. Time in hh:mm. Scale bar: 10 μ m.

Figure L4: Perturbation of actin organization in MDCK cysts using latrunculin A. Representative images show the cysts before treatment (top panel) and after latrunculin A treatment (bottom panel), highlighting the effects of actin disruption on cyst morphology and lumen structure. Scale bar: 10 μ m.

Reviewer #3 (Remarks to the Author):

The authors have revised their manuscript by including a number of new experiments, including a comparison of hydrostatic pressures across different cell systems and micropipette inflation experiments. Figure L16 shows an elegant way to create lumens in cell cords, and this Figure must be included in the final manuscript (and not just shown for the referees).

Thank you for the positive comment. The "cell cord" Figure is already included in the main Figure 1 (panels f and g).

The surface tension measurement, however, is not addressing ref. 3 concern that a reduction of luminal surface tension is a key driver, as the authors measure surface tension at the basolateral rather than luminal, apical cell surface.

We now specify in the new version of the MS p. 10 that we aspirated the full monolayer which allowed us to measure the surface tension of the whole monolayer.

If the authors include L16 in the published paper, the manuscript is fine.

It is added.

Reviewer #4 (Remarks to the Author):

In the revised manuscript, the authors have been partially responsive to my concerns. My sense is that the manuscript is still a bit descriptive and less mechanistic; however, there are lot of interesting experiments and analyses here, and the study of lumen dynamics, particularly fusion, is novel and will be of interest to the field. So, I generally support publication.

We thank the reviewer for the positive comment.

However, the following points should be addressed.

1. As mentioned in previous review, the term epiblast is not appropriate here – it makes it seem as if live epiblasts are being studied. The references cited from 2014 and 2017 supporting use of this term do not reflect the current standard and terminology of the community. Perhaps epiblast model, epiblast organoid, or synthetic epiblast would be more appropriate.

Thank you for this suggestion. We changed for epiblast model in several places in the text and kept also epiblast for simplicity. We explain this usage in the paragraph "Cell sources and expansion".

2. The authors make the claim that the cell convergent dynamics in the epiblast model leads to fusion, and have supported this claim with the blebbistatin perturbation (Fig. 8). I don't think this is the strongest experiment, given that adding blebbistatin is a blunt perturbation, but its ok for this experiment. However, the authors should support this at least with quantification of the experimental phenomena (centripetal movement and fusion rate) rather than simply show images.

To address this comment, we have added within this Fig. 8 measurements of centripetal movement and fusion rate, as shown in Figure L5 (right panel below) in the presence and in the absence of blebbistatin. This analysis confirms that centripetal motion is inhibited when myosin is not active.

Figure L5: Perturbation of lumen fusion in the epiblast in the presence of blebbistatin. (a) Representative images of the lumen fusion process in the epiblast. The top panel shows the control condition treated with DMSO, and the bottom panel shows the effect of blebbistatin treatment. Lumens are visualized with SiR-actin (gray). Scale bar: $5\mu\text{m}$. Right panel: Single-cell trajectories during the fusion process under DMSO treatment (top) and blebbistatin treatment (bottom). Time is indicated by the color bar, with the red point marking the last time point.

3. The origins of lumen nucleation in epiblasts (lines 184 – 190) were not clear to me. Why are the authors arguing that lower lumen pressure leads to this alternate form of lumen nucleation? Is this simply a matter of cell division growth rate and geometrically fitting the growing cells within a spherical structure? The proposed mechanistic insight should be explained clearly here.

Unlike MDCK cysts, lumen of epiblast organoids does not nucleate spontaneously after cell division because the luminal pressure is too low, but apical molecules are expressed between two cells. The cells are polarized by the apical proteins and ECM. In cells with polarity, the rosette structure is energetically more favorable. Nevertheless, once rosette is created and if the cell number exceed about 10 cells, total force by the contractile tension of cell lateral walls becomes strong, and even if the luminal pressure is almost 0 [Pa], it overcomes the contractile force of the apical membrane and opens the lumen at the centre of the rosette. This is equivalent to minimizing the surface energy of total cell-cell and cell-lumen interfaces. This is shown in terms of free energy in Figure 6 of the revised manuscript together with more explanations and new schemes in Fig. 6e-f.

The apical and the lateral sides generate a total force that needs to encompass the resisting force associated to the cell monolayer and the Matrix. As a consequence, more cells are needed to outcompete this threshold force. The computation shows that 10 cells are sufficient to be in this situation and open a lumen in the absence of hydrostatic pressure. In contrast, high hydrostatic pressure spontaneously allows this opening of lumen already with 2 cells.

This is explained p. 5 and in details in Suppl. Note.

4. The abstract notes that “We finally use MDCK cysts to manipulate cell adhesion and lumen volume and we successfully reproduce the fusion dynamics of pancreatic spheres and epiblasts”. It is not clear to me where the MDCK cysts are manipulated to induce the centripetal motion to induce fusion, to model the fusion dynamics of the epiblast model. This should be clarified/quantified, or the claim should be removed.

We have now included the centripetal motion to induce fusion in Figure Suppl. 11 in addition to other tests reported in Fig. 7 and Fig. 8. We also add now quantification for this motion next to the centripetal motion data across systems.

Figure L5. Lumen fusion comparison and aspect ratio analysis of MDCK WT cysts, ZO-KO cysts, and epiblasts. The left panel shows lumen fusion behavior in MDCK WT cysts, ZO-KO cysts, and epiblasts. ZO-KO mutants display fusion dynamics similar to epiblasts. MDCK WT and ZO-KO cysts are labeled with Podocalyxin (red) and E-cadherin (green), while epiblasts are stained with SiR-actin (grey). The right panel presents a quantitative analysis of the aspect ratio over time, where the aspect ratio is defined as the ratio of long axis to short axis; ratio 1 (in red) corresponds to a spherical shape. Scale bar: 10 μm .

5. At a broader level and related to 4, it still seems to me that the epiblast model behavior is fundamentally different than the behavior of the pancreatic sphere and MDCK cyst (as is also suggested by their final takeaway figure 9). The nucleation behavior, pressurization of the lumen, and lumen fusion dynamics are all quite different, yet the authors are trying to put these different behaviors into a universal framework. I don't think this is really supported by the data, and I don't think it takes away from the paper if they discover two modes of lumen nucleation and fusion. However, I would give the authors some latitude here in what they conclude, but perhaps they could more clearly explain the basis for their conclusion of universality in the discussion.

The nucleation and growth of lumens are governed by the same rule in three systems, which is the balance between the luminal pressure and the contraction of the cell lateral walls and apical membrane, and this is consistent with the minimization of the total free energy. Epiblast organoid cannot spontaneously nucleate a lumen during cell division because of low luminal pressure, which costs more free energy. In the process of minimising free energy, the cell passes through a low free energy rosette state. This is achieved by using the active fluctuations (centripetal motion) of the cells. However, as explained in Figure L6, even when the internal pressure is nearly 0 Pa, a rosette consisting of more than 10 cells is unstable, and leads to the formation of a lumen that further lowers free energy. In MDCK cyst and pancreas spheres, lumens spontaneously nucleate between two cells due to a finite luminal pressure. Lumen fusion process is also known as the result of free energy minimisation under the influence of active fluctuations of cell motion. In this sense, these systems follow the universal rules.

This is added in the new Fig. 6 in the revised manuscript and in the discussion p. 7.

Figure L6. A schematic illustration of lumenogenesis with and without hydrostatic pressure. a. Top : schematic representation of the force balance at vertex, highlighting the lateral (lat) and apical forces (ap) (top). Bottom: the schematics depicts the Young-Laplace law with γ the surface tension and p the pressure (p_{ECM} , the extracellular matrix pressure, and p_{LM} luminal pressure). b. Simplified schematic of the model for lumenogenesis in the absence of hydrostatic pressure. When Δp is close to 0 Pa, lumen formation requires a minimum threshold of 10 cells to nucleate a lumen. In contrast when $\Delta p \gg 0$ Pa, 2 cells spontaneously nucleate a lumen.

** See Nature Portfolio's author and referees' website at www.nature.com/authors for information about policies, services and author benefits.

This email has been sent through the Springer Nature Tracking System NY-610A-NPG&MTS

Confidentiality Statement:

This e-mail is confidential and subject to copyright. Any unauthorised use or disclosure of its contents is prohibited. If you have received this email in error please notify our Manuscript Tracking System Helpdesk team at <http://platformsupport.nature.com> .

Details of the confidentiality and pre-publicity policy may be found here <http://www.nature.com/authors/policies/confidentiality.html>

Privacy Policy | Update Profile